# Analysis of Poplar's (*Populus nigra* ita.) Root Systems for Quantifying Bio-Engineering Measures in New Zealand Pastoral Hill Country

Ha My Ngo [1,2,*], Feiko Bernard van Zadelhoff [2,3], Ivo Gasparini [2], Julien Plaschy [2], Gianluca Flepp [2], Luuk Dorren [2], Chris Phillips [4], Filippo Giadrossich [1] and Massimiliano Schwarz [2]

1   Department of Agriculture, University of Sassari, Viale Italia 39A, 07100 Sassari, Italy; fgiadrossich@uniss.it
2   HAFL, Bern University of Applied Sciences, Länggasse 85, CH-3052 Zollikofen, Switzerland; massimiliano.schwarz@bfh.ch (M.S.)
3   Institute of Geography (GIUB), University of Bern, CH-3012 Bern, Switzerland
4   Manaaki Whenua-Landcare Research, Lincoln 7608, New Zealand
*   Correspondence: hmngo@uniss.it

**Abstract:** *Populus nigra* ita. is an important tree species for preventing rainfall-triggered shallow landslides and hydraulic bank erosion in New Zealand. However, the quantification of its spatial root distribution and reinforcement remains challenging. The objective of this study is to calibrate and validate models for the spatial upscaling of root distribution and root reinforcement. The data were collected in a 26-year-old "Tasman" poplar stand at Ballantrae Hill Country Research Station in New Zealand. We assessed root distribution at different distances from the stem of four poplar trees and from eleven soil pits along a transect located in a sparse to densely planting poplar stand. 124 laboratory tensile tests and 66 field pullout tests on roots with diameters up to 0.04 m were carried out to estimate root mechanical properties. The results show that the spatial distribution of roots can be well predicted in trenches of individual tree root systems ($R^2 = 0.78$), whereas it tends to overestimate root distribution when planting density was higher than 200 stems per hectare. The root reinforcement is underestimated within single tree root systems ($R^2 = 0.64$), but it performs better for the data along the transect. In conclusion, our study provided a unique and detailed database for quantifying root distribution and reinforcement of poplars on a hillslope. The implementation of these models for the simulation of shallow landslides and hydraulic bank erosion is crucial for identifying hazardous zones and for the prioritization of bio-engineering measures in New Zealand catchments. Results from this study are useful in formulating a general guideline for the planning of bio-engineering measures considering the temporal dynamics of poplar's growth and their effectiveness in sediment and erosion control.

**Keywords:** root reinforcement model; root distribution model; root bundle model; shallow landslides; poplar; forestry management

## 1. Introduction

Vegetation is a key factor for the control of soil erosion and sediment in many environmental systems [1–5]. In New Zealand, this was recognised in the 19th century and in 1941 when the Soil Conservation and Rivers Control Act was adopted [6,7]. The country's hilly pastures represent one of the most susceptible landscapes to erosion, resulting in a significant issue for on-site and off-site effects such as loss of pasture productivity and declining water quality. Most sediment is mobilized by processes such as shallow landslides and hydraulic bank erosion where vegetation has a strong influence in reducing their magnitude and frequency [8,9].

In the case of shallow landslides, vegetation contributes to slope stability mostly by increasing soil strength through root reinforcement in the triggering area [10–23]. Additionally, vegetation reduces the probability of the occurrence of hydrological triggering

conditions through the regulation of the water balance within the potential landslide area and the hydrological contributing area (e.g., through interception of precipitation by leaves, effects of evapotranspiration on soil water content, increase in soil porosity and permeability, and act as hillslope-scale preferential flow paths [19,24,25]). Shallow landslides in most cases are triggered by rainfall events that lead to the build-up of pore water pressure in the soil and increase soil weight while reducing its shear strength [17,19]. As a consequence, the soil mass fails when the balance between the stabilizing and destabilizing forces is lost.

Root reinforcement is determined by both mechanical properties and the distribution of roots, and thus depends greatly on tree species and stand density. There are three different mechanisms of root reinforcement [17,19]:

- Lateral root reinforcement: takes place at the transition between the sliding surface and the stable surface and is significantly affected by the type of soil deformation, root density, and spatial distribution of the root system.
- Basal root reinforcement: occurs when roots cross the shearing surface and thus depends on the shearing surface depth. It is important for shallow landslides but negligible for deep-seated landslides.
- Stiffening of the soil mass: increases the stability through root buttressing and arching [11].

In the case of hydraulic bank erosion, roots increase the value of critical shear strength of soil along the banks [26], thereby reducing hydraulic bank erosion. The effectiveness of this mechanism strongly depends on the type of soil, the intensity and duration of the water discharge, and the amount and distribution of roots. The root parameter that is usually implemented in models is the root area ratio (RAR) [27]. For this reason, quantifying the spatio-temporal distribution of RAR depending on tree species and location is an important step in the implementation of root effects in bank stability models.

New Zealand's pastoral hill country is a good example of how anthropogenic land use has altered geomorphological processes. Specifically, the extension of natural forests was reduced from about 85% prior to human habitation to 29% of land area nowadays [28,29]. Most of the pastoral hill country is used today for farming, grazing (mainly sheep and cows, and also deer). Within a few decades after the first arrival of Europeans in the middle of the 19th century, the extensive forest clearance and conversion to pasture applied by the colonists lead to a dramatic increase of erosion [6,30,31]. Landslides have caused considerable loss of productive soil [32,33], water holding capacity, movement of sediment in streams and rivers, and decline in water quality [34,35]. Studies have documented that soil productivity recovery from shallow landslide affected areas takes a long time (decades) and annual pastoral production is less than 80% of that of areas unaffected by landslides [9,36,37].

Exotic fast-growing trees such as pines (*Pinus radiata*), willows (*Salix* spp.), and poplars (*Populus* spp.) have been widely introduced on pastoral hill country because of economic benefits including soil conservation, serve as fodder, shelter, and shade for livestock, lower wind speed, produce carbon credits, and generate timber production [31,38,39]. For example, shade belts in hot weather have been declared to boost reproductive performance and growth rates while decreasing the risk of hypothermia and death in lambs [40]. Indigenous species with a slower growth rate were not used in problematic areas [41]. The most suitable and proven tree species were poplar (*Populus* spp.) and willow (*Salix* spp.) because they are (i) easy to reproduce vegetatively by means of cuttings, (ii) readily established from large poles in the presence of stock, (iii) easily transported and can be planted on steep hill country, (iv) grow quickly with 1 to 4 m per year in the first years, (v) tolerate wet soil conditions during long periods and do not affect pasture growth unless planted at narrow spacing or high density, (vi) possess an extensive and strong root system which is able to rapidly stabilize the soil mass [8,41,42], (vii) obtain a high evapotranspiration rate so can remove a large amount of water from soil, (viii) obtain good capacity to recover from mechanical damage such as soil movements or stock impacts [41], (ix) provide shade,

shelter, and quality fodder (especially during drought periods), (x) sequester carbon [43,44], and lastly, (xi) may provide income for farmers with their timber.

In the present study, we focused on the effects of root distribution of *Populus deltoides x Populus nigra* "Tasman" poplar on pastoral hill country in New Zealand. "Tasman" poplar is one of the two clones of the most common poplar hybrid in New Zealand [45]. "Tasman" poplar, as well as "Veronese" poplar, are hybrids of the American (*Populus deltoides*) and the European (*Populus nigra*). However, the former is a male clone while the latter is a female one [46]. Tasman poplar has a narrower crown and acquires more water compared to Veronese poplar but is more resistant to rust. The incredible growth rate of these *Populus deltoides x P. nigra* hybrids was highlighted by McIvor et al. [47]. Although some studies have quantified the spatial distribution of roots in young trees [8,33,45], data and modeling for the quantification of older trees are missing. The present study aims to fulfill this research gap. Specifically, the objectives are to quantify the spatial root distribution and root reinforcement of 26-year-old Tasman poplars and apply analytical models for upscaling and implementing root reinforcement in geomorphological models.

## 2. Materials and Methods

### 2.1. Site Description

Data were collected at Ballantrae Hill Country Research Station (40°18′57″ S, 175°50′24″ E) located in the Manawatu region, in the south of North Island (Figure 1). The elevation of the site is ca. 130 m ASL (Manaaki Whenua—Landcare Research 2019) and the terrain is slightly northeast facing. The climate is temperate: frosty winters and cool summers with the maximum 30 °C on exceptional occasions. Details of our study site are shown in Table 1.

**Table 1.** Summary of study site characteristics (data from the LRIS Portal).

| Variables | Description | Variables | Description |
| --- | --- | --- | --- |
| Region | Hard Rock Hill Country | Soil pH | 5.5–7.5 |
| Province | Eastern Soft Rock | Mean Erosion rate | 2604 t/km$^2$/yr |
| Rock type | Sedimentary rocks | Mean annual soil temperature | 11–15 °C |
| Soil texture | Silt loam | Topsoil gravel content | 0–4 % |

The site is steep to very steep developed on siltstone and banded mudstone, which are very loose to compact, with moderate to high (more than 1200 mm) rainfall events. Shallow landslides are the dominant erosion form and 1–10% of the area is affected by earthflows. The erosion rate recorded in 2001 was ca. 2604 t/km$^2$/yr. Soil pH ranged from 5.5 to 7.5. Potential maximum rooting depth is moderately deep −0.6 to 1.19 m belowground.

The poplar stand was 26 years old at the time when field measurements were made. The poles were distributed in a radial pattern (Nelder layout) (Figure 2). Tree spacing was about 3.5 m, corresponding to approx. 800 stems per hectare (sph) in the densest zone and up to 12.9 m (ca. 60 sph) in the sparsest one. Double circular trenches were excavated around four trees located in the sparse zone to recorded data on root distribution and root mechanical properties (Figure 2). The high soil clay content explains the low vertical permeability of the soil and the hydro-morphic characteristics (Figure 3). A total of 11 soil pits were also dug to quantify the effect of stand density on the root distribution of overlapping root systems.

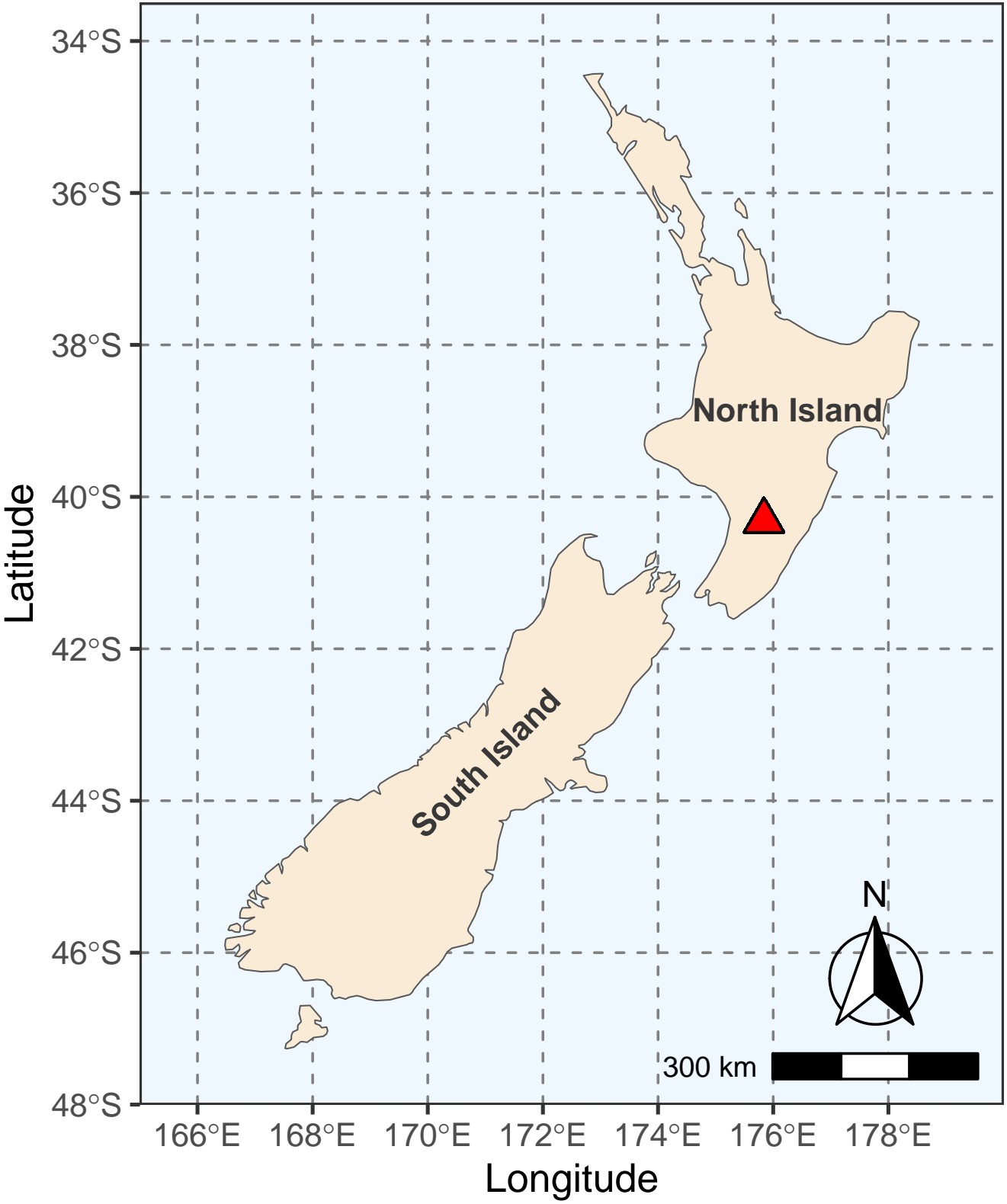

**Figure 1.** Location of the study site (presented as the red triangle) in New Zealand.

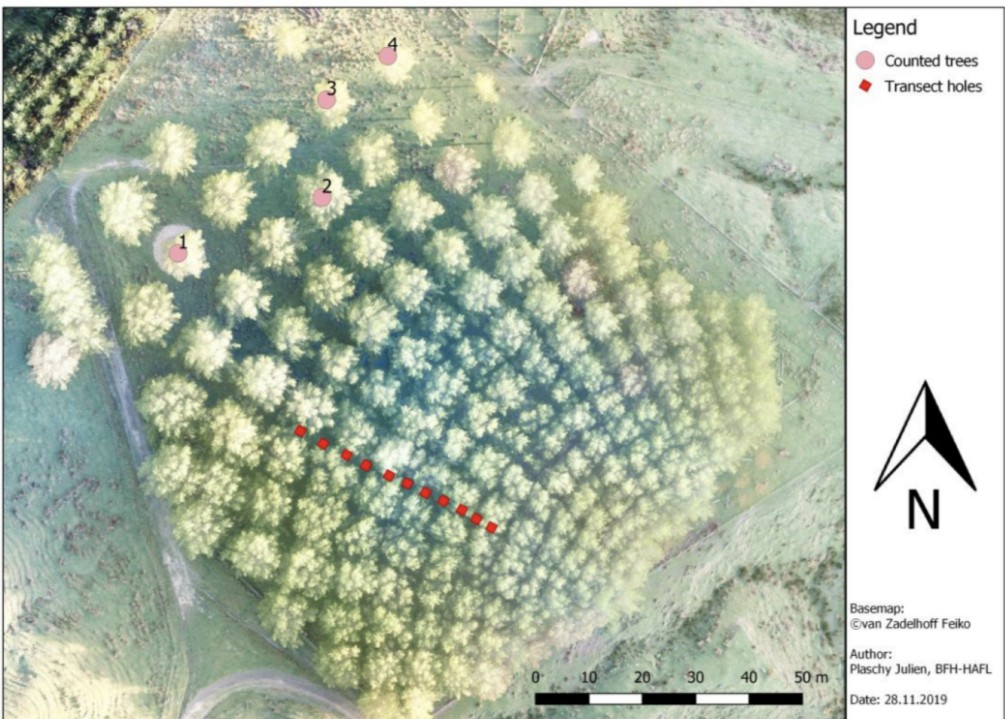

**Figure 2.** Overview of the tree stand, being a "nelder" planting trial. Trees from the wider spaced part of the stand were chosen to have less overlap between the root systems of neighbouring trees and are represented by pink dots. Transect pits are represented as red squares.

Root counting and measurement were conducted manually from the 30 September–8 November 2019. The study was divided into two parts with different objectives. Details on the methodologies are presented in the following sections.

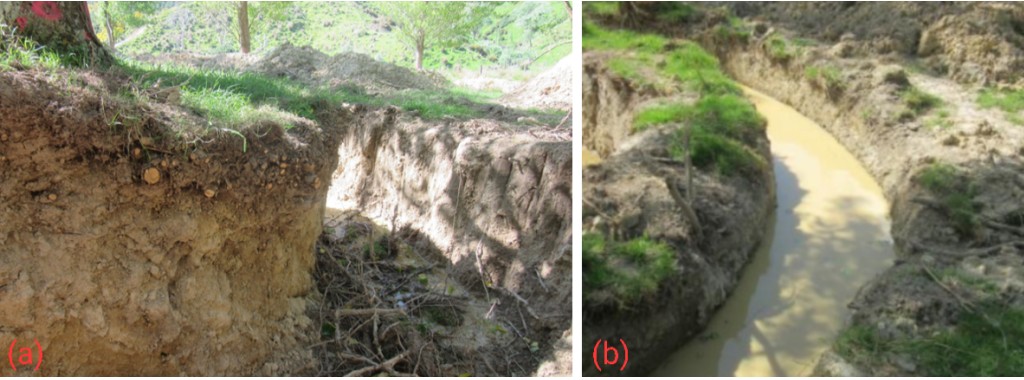

**Figure 3.** One of the trenches (**a**) before a rainfall event and (**b**) with more than 0.7 m water after 3 days without rain, indicating slow soil drainage.

### 2.2. Root Distribution Measurements

#### 2.2.1. Individual Tree Measurements

To quantify the spatial root distribution at the single tree scale, we used the method similar to Giadrossich et al. [48]. We selected four trees (Tree 1, 2, 3, and 4) in the sparsest part of the stand in order to minimize the influence from neighbouring trees, as shown in Figure 2, with DBH (Diameter at Breast Height) of 0.41, 0.42, 0.51, and 0.56 m, respectively. Two 360° trenches around the stem, with a width and depth of approximately 1 m, were dug with an excavator. The distances from the middle of the tree stem to these trenches were 1.5 m, 2.5 m, 3.5 m, and 4.5 m. Trenches were divided into eight 45° sectors. Sectors 1 and 8 were situated uphill. Each sector was separated into 7 depth layers [0–0.15 m],

[0.15–0.30 m], [0.30–0.45 m], [0.45–0.60 m], [0.60–0.75 m], [0.75–0.90 m], and [0.90–1.05 m]. Living fine roots belonging to the sampled tree were counted and assigned to diameter class [0.5–1.5 mm], whereas living coarse roots were classified into 1 mm diameter classes [1.5–2.5 mm], [2.5–3.5 mm], [3.5–4.5 mm], and so on. The maximum recorded root diameter was 40 mm.

### 2.2.2. Transect Measurements

To validate the upscaling of the model at the stand scale, eleven square pits were dug in the stand along a transect between two rows of trees as shown in Figure 2. Each pit had four soil faces, 1 m wide and 0.9 m deep. Roots were counted and recorded separately for each face (1–4) of each square pit in the same manner as for the circular trenches. All the surrounding trees' parameters within a distance of 30 times its DBH to the pit were recorded as they possibly affected the presence of roots in the pit. The transect pits are numbered from the sparsest zone (1) to the densest zone (11) (Table 2). The tree density around the pits was approximately 84 stems per hectare (sph) in the sparse zone and 770 sph in the dense zone.

**Table 2.** Tree distribution surrounding transect's pits. DBH refers to tree diameters measured at breast height; dist. is the distance of pit-tree.

| Transect Pit | No of Surrounding Trees | Mean Dist. [m] | Min Dist. [m] | Max Dist. [m] | Average DBH [m] | DBH at Min Dist. [m] |
|---|---|---|---|---|---|---|
| 1 | 8 | 10.5 ± 4.0 | 4.6 | 16 | 0.58 | 0.52 |
| 2 | 10 | 11.5 ± 4.2 | 6.2 | 16.3 | 0.57 | 0.57 |
| 3 | 11 | 10.5 ± 4.7 | 3.2 | 16.8 | 0.54 | 0.55 |
| 4 | 12 | 11.0 ± 3.7 | 5.9 | 14.5 | 0.53 | 0.54 |
| 5 | 11 | 10.2 ± 3.8 | 4.1 | 15.3 | 0.53 | 0.55 |
| 6 | 14 | 10.8 ± 3.8 | 5.1 | 17.1 | 0.52 | 0.46 |
| 7 | 15 | 10.5 ± 3.8 | 3.5 | 17.1 | 0.51 | 0.46 |
| 8 | 14 | 9.9 ± 3.7 | 4.8 | 15.9 | 0.51 | 0.46 |
| 9 | 15 | 9.8 ± 3.8 | 3.1 | 15.5 | 0.49 | 0.52 |
| 10 | 14 | 8.9 ± 3.1 | 4.5 | 12.7 | 0.48 | 0.47 |
| 11 | 16 | 9.0 ± 3.7 | 2.7 | 15.9 | 0.47 | 0.47 |

### 2.2.3. Root Area Ratio

The cross-sectional area of roots per area of soil profile, known as the root-area-ratio (RAR), is calculated for all trenches and transect's pits. The RAR is defined as

$$RAR = \sum_{i=1}^{n} \frac{A_{r,i}}{A_s} \qquad (1)$$

where $A_r$ is root cross section area [m$^2$] and $A_s$ is soil area [m$^2$], $i$ is the root diameter class, and $n$ is the number of root diameter classes.

### 2.3. Root Pullout Tests

The mechanical properties of the root-soil interaction were tested by field pullout experiments as described in previous studies [15,18,20,49,50]. Field root pullout testing is considered the most representative method to quantify root reinforcement when using the Root Bundle Model approach (RBMw) [51]. Trees used for the pullout tests were all in good health and were of similar DBH (0.4 m). Selected roots were carefully excavated to expose a sufficient length, anchored by threaded rods, and pulled towards the tree stem by a pullout device. The pullout device consists of an aluminium frame equipped with a steel rope winch and a crank handle. Force applied on the roots was measured by a load cell

with a nominal maximum load capacity of 2 t. A total of 66 pullout tests and 124 laboratory tensile tests were performed.

### 2.4. Root Distribution Modeling

Root distribution is modeled using the Root Distribution Model (RDM) described by Schwarz et al. [52]. The RDM estimates the number of roots in diameter class size *i* that cross a 1 m width vertical soil profile at a distance *d* from an isolated tree stem with the diameter at breast height (DBH, in [m]) $\phi_t$ following the equation:

$$N_{i,t}(d,\phi_t) = \begin{cases} D_{fr}\frac{[ln(1+\phi_{max})-ln(1+\phi_i)]}{ln(1+\phi_{max})}\left(\frac{\phi_i}{\phi_0}\right)^\beta, & \text{with } d < d_{max} \text{ and } \phi_i < \phi_{max} \\ 0, & \text{otherwise} \end{cases} \qquad (2)$$

$d_{max}$ is the maximum rooting distance from the stem [m], $D_{fr}$ is the density of fine roots (smaller than 1.5 mm) per horizontal meter, $\phi_i$ is the mean root diameter in class size i [m], $\phi_0$ is a reference diameter (in this paper equal to 1 mm), $\phi_{max}$ is the maximum root diameter [m], and $\beta$ is a constant exponent.

$$d_{max}(\phi_t) = \psi\phi_t \qquad (3)$$

$$\phi_{max} = \frac{d_{max} - d}{\eta} \qquad (4)$$

$\psi$ is a proportionality constant, $\phi_t$ is the tree diameter at the breast height [m], and $\eta$ is a dimensionless scaling coefficient.

The density of fine roots [0.5–1.5 mm] crossing a 0.9 m depth and 1 m width vertical soil profile at a given distance *d* from an isolated tree stem with DBH $\phi_t$ is calculated as:

$$D_{fr}(d,\phi_t) = \left[\frac{\mu(\phi_t^2\frac{\pi}{4})}{d_{max}2\pi d}\right]\left(\frac{d_{max}-d}{d_{max}}\right), \text{ with } d < d_{max} \qquad (5)$$

where $\mu$ is the pipe coefficient.

Model parameters $\mu$, $\beta$, $\psi$, and $\eta$ were calibrated by minimizing the Sum of Squares Errors (SSE) obtained as the difference between modeled and measured root distribution data.

The RDM simulates lateral root distribution only, without considering the vertical distribution. The calibration of the RDM is particularly useful for the application of models such as BankforNET [26,27,53], and for the modeling of hydraulic bank erosion influenced by riparian vegetation.

### 2.5. Root Reinforcement Calculations: From Single Root to Root System
#### 2.5.1. RBMw

The Root Bundle Model with Weibull survival function (RBMw) proposed in [15,17,51] is used to quantify the reinforcement due to a root bundle. Root tensile force as a function of displacement and root distribution are two essential inputs of the model. Applying in-situ data allows a better fit of the model calibration of the local field conditions [49], assuming that the pullout force of a single root is not affected by neighbouring roots [54]. A power-law relationship is used to fit the regression curve between maximum tensile force and root diameter:

$$F_{max}(\phi) = C(\phi)F_0\phi^\alpha \qquad (6)$$

$F_{max}$ is the maximum tensile force [N], $\phi$ is root diameter [m], $F_0$ is a constant, and $\alpha$ is an exponential parameter. Because the fitting curve overestimates the strength of roots with a diameter smaller than 5 mm in some cases, a cumulative normal distribution function (Equation (7)) with values ranging from 0 to 1 is used to improve the model fit [55].

$$C(\phi) = \frac{1}{2}[1 + erf(\frac{\phi - \phi_m}{\phi_{sd}\sqrt{2}})] \qquad (7)$$

where $\phi_m$ and $\phi_{sd}$ are coefficients corresponding to the mean and standard deviation of the cumulative normal distribution.

An apparent secant spring constant was calculated by the ratio of maximum root pullout force over the displacement at root failure.

$$k = k_0 \phi \tag{8}$$

where $k$ is the spring constant [N/m], $\phi$ is the root diameter [m], and $k_0$ is a root spring constant scaling factor.

In the RBMw, the survival probability of each root diameter is calculated as a function of the normalized displacement, $\Delta x^*(\phi)$.

$$\Delta x^*(\phi) = \frac{\Delta x}{\Delta x_{max}^{fit}(\phi)} \tag{9}$$

The Weibull survival function is defined as below:

$$S(\Delta x^*) = exp[-(\frac{\Delta x^*}{\lambda})^\omega] \tag{10}$$

where $\omega$ is the Weibull shape factor and $\lambda$ is the Weibull scaling factor. The parameters $k_0$, $\alpha$, $\lambda$, and $\omega$ are calibrated using measured data from field pullout tests and tensile tests, by minimizing the Sum of Squared Errors (SSE).

The total root reinforcement of a root bundle is calculated as the sum of all tensile forces of roots in the bundle at different displacements. Lateral tensile root reinforcement is expressed in N/m, considering all the roots crossing a 1 m width vertical soil trench.

$$RR_{bundle}(\Delta x) = \sum_{\phi=1}^{\phi_{max}} n_\phi F(\phi_i, \Delta x) S(\Delta x_\phi^*) \tag{11}$$

The lateral root reinforcement of all soil trenches in the study area is calculated using the calibrated RBMw.

### 2.5.2. Root Reinforcement at the Root System Scale

The maximum lateral root reinforcement $RR_{max}$, defined as the peak of the force-displacement curve resulting from the RBMw, is used to upscale the value of root reinforcement from a single root to the root system scale. $RR_{max}$ is calculated as a function of the tree DBH, $\phi_t$ [m], and the distance from the tree stem, $d$ [m]. The function for the lateral reinforcement is assumed to follow the gamma density distribution $\Gamma$ [15,55] as below:

$$RR_{max}(\phi_t, d) = \begin{cases} a \cdot \phi_t \cdot \Gamma(\frac{d}{\phi_t}, b, c), & \text{for } d < 18.5 \cdot \phi_t \\ 0, & \text{for } d \geq 18.5 \cdot \phi_t \end{cases} \tag{12}$$

where $\phi_t$ is tree size (DBH, in [m]), $d$ is the distance from the tree stem [m], $a$ is scaling factor, $b$ is shape parameter, and $c$ is rate parameter [25]. The calibration of the parameters $a$, $b$, and $c$ is achieved by minimizing the Sum of Squared Errors.

From the lateral root reinforcement $RR_{max}$, it is possible to calculate the basal reinforcement in accordance with Gehring et al. [55] as below:

$$RR_{basal}(z) = RR_{max} \cdot \Gamma(z, z_\alpha, z_\beta) \tag{13}$$

where $z$ is the soil depth, $z_\alpha$ and $z_\beta$ are calibrated coefficients of the gamma density function using minimizing SSE.

This model does not take spatially differentiated root distribution along the slope gradient around a tree stem. It assumes that the roots are symmetrically distributed all around the tree [15,56].

*2.6. Statistic Analyses and Models Validation*

RDM, RBMw, and root reinforcement models were implemented in the open-source programming language R software. Cross-validation [57–60] is used to estimate the generalization performance and to evaluate the proposed models. Cross-validation has been widely regarded as a standard method in model evaluation and selection [61,62]. The data is split into train/test in the percentage ratio of 80/20. This partition has been commonly applied in various research fields for splitting data and independent accuracy assessment because it was proved to perform better for large datasets, avoid overfitting, achieve the best result as well as decrease computational time [63–67]. We then evaluated model performance through the Sum of Squares Errors (SSE), $R^2$, and the difference in normalised SSE between training and testing datasets to examine the model stability. This operation is repeated 30 times, changing training and testing datasets randomly, in order to prove the convergence of the results. The formula is similar to the Mean Bias Error (MBE) as below:

$$\overline{Var} = \frac{1}{n} \sum_{i=1}^{n} (N\_SSE_{train,i} - N\_SSE_{test,i}) \tag{14}$$

where $N\_SSE_{train,i}$ is normalised SSE of the training dataset, $N\_SSE_{test,i}$ is normalised SSE of testing dataset, $n$ is the number of assessed loops, and $\overline{Var}$ is the mean variance between the two datasets.

**3. Results**

*3.1. Root Distribution Measurements and Modeling*

3.1.1. Root Distribution Measurements of Single Root Systems

The number of measured fine roots [0.5–1.5 mm] decreases with soil depth and also with the increase in distances from the stem in all trenches (Figure 4). Considering horizontal distribution, the greatest number of fine roots is recorded in the 1.5 m trench in all trees.

The largest changes in fine root frequency are observed between the 1.5 m and 2.5 m trenches around all trees, whereas the number of fine roots in the three farther distances does not change remarkably. In the first soil layer, the reduction of fine roots between the 1.5 m and 2.5 m trench of tree 1, 2, 3, and 4 are 61%, 61%, 51%, and 59%, respectively.

Considering the vertical root distribution, the number of fine roots is reduced dramatically by increasing soil depth. In the first trench, the total number of fine roots in the first soil layer is between 136 roots/m to 219 roots/m, decreasing down to 37 roots/m to 52 roots/m in the second layer. Tree 3 exhibits the most fine roots in the first soil layer, however, fewer fine roots are present in deeper soil layers compared to other trees. Although the number of fine roots of the largest tree is the least in the nearest trench, it becomes the greatest in the farthest trench. In particular, fine roots of larger trees reaches deeper soil layers and expands further than smaller ones.

The amount of coarse roots has a similar pattern as fine roots, with decreasing trend from 1.5 m to 4.5 m distances (Figure 5). The majority of coarse roots is in the first 0.15 m soil layer. Tree 4 has the greatest number of coarse roots in both horizontal and vertical directions. In the horizontal direction, the number of coarse roots in tree 4 in the 4.5 m trench (presented as purple bars) is twice that of tree 1 and tree 2. There are huge gaps in the number of coarse roots from the 1.5 m to the 2.5 m trench in all trees, which reflected the pattern measured in the fine root distribution. Given coarse root density in the 0–0.15 m soil layer, the differences between the 1.5 m and 2.5 m trench in tree 1, 2, 3, and 4 are recorded to be 54%, 54%, 56%, and 56%, respectively.

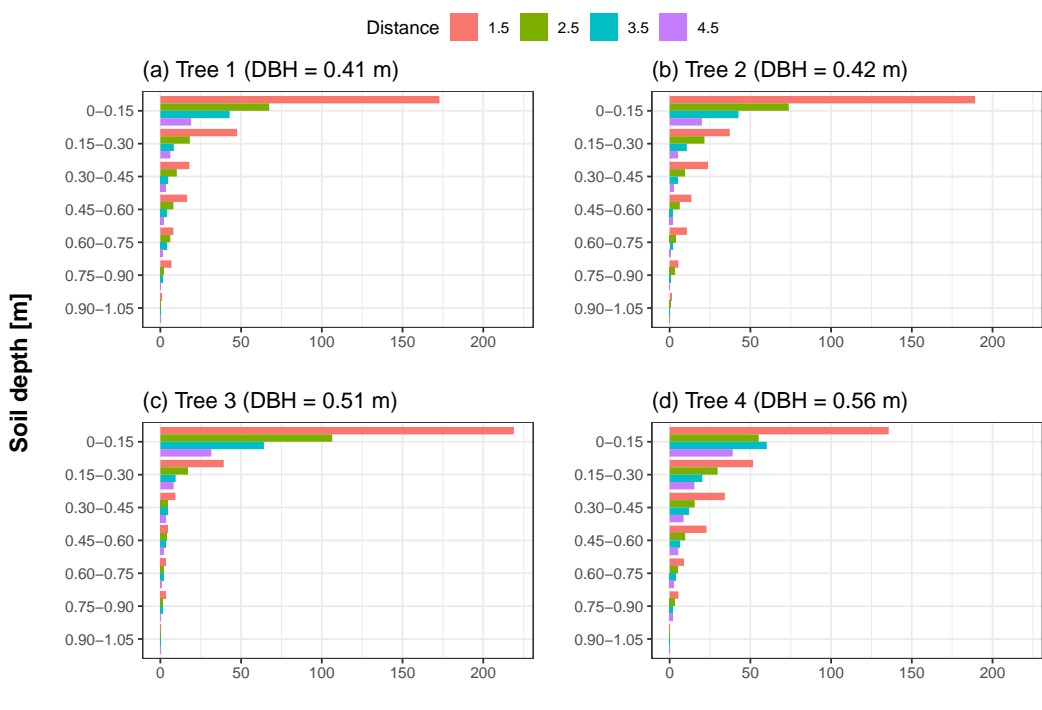

**Figure 4.** Mean measured number of fine roots [0.5–1.5 mm] of "Tasman" poplars in each soil depth of 1 m width at different distances 1.5 m (red color), 2.5 m (green color), 3.5 m (blue color), and 4.5 m (purple color) from stem with different DBH (**a**) 0.41 m, (**b**) 0.42 m, (**c**) 0.51 m, and (**d**) 0.56 m.

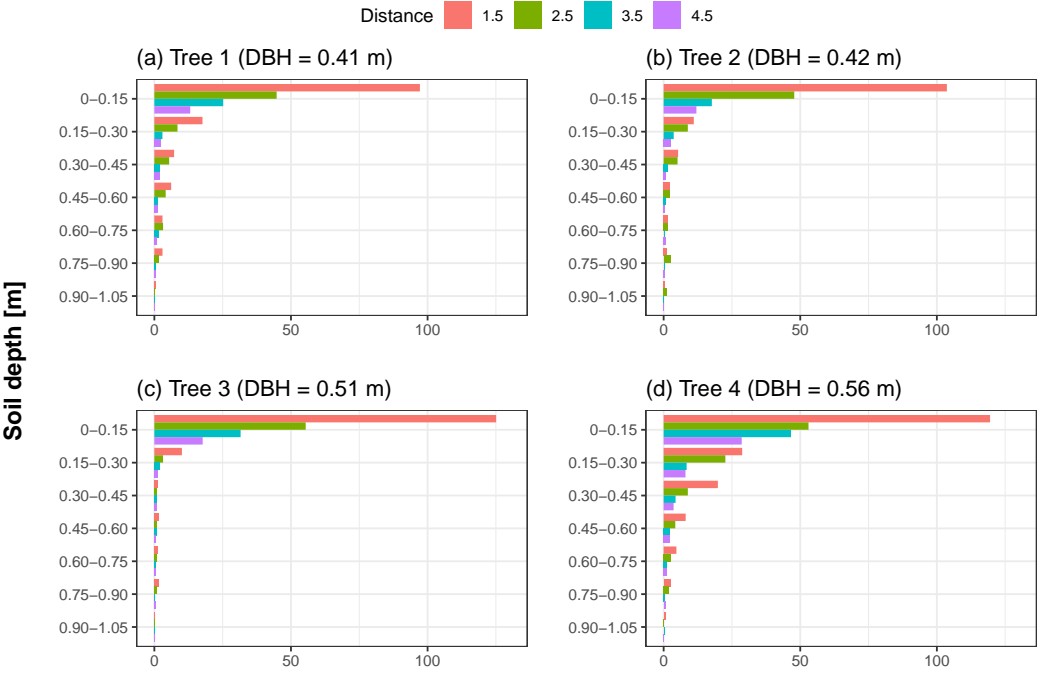

**Figure 5.** Measured number of coarse roots (>1.5 mm) of "Tasman" poplars in each soil depth at different distances (1.5 m (red color), 2.5 m (green color), 3.5 m (blue color), and 4.5 m (purple color)) from the stem with different DBH (**a**) 0.41 m, (**b**) 0.42 m, (**c**) 0.51 m, and (**d**) 0.56 m

Table 3 presents the measured partition of three root categories: fine roots, 1.5–10.5 mm roots and >10.5 mm roots at the first two soil layers and two nearest distances from the

tree stem. Overall, fine roots comprise the greatest proportion of the total number of roots in all trees, followed by 1.5–10.5 mm root classes and roots >10.5 mm.

**Table 3.** Measured composition of root classes: fine roots, 2–10 mm root class and >10.5 mm roots at the first two soil depths and first two distances from the tree stem.

| | | **1.5 m Distance** | | | **2.5 m Distance** | | |
|---|---|---|---|---|---|---|---|
| **Tree** | **Depth** | **% Fine Roots** | **% 1.5–10.5 mm Roots** | **% Roots > 10.5 mm** | **% Fine Roots** | **% 1.5–10.5 mm Roots** | **% Roots > 10.5 mm** |
| 1 | 0–0.15 | 64 | 33 | 3 | 60 | 36 | 4 |
| 1 | 0.15–0.3 | 73 | 25 | 2 | 68 | 29 | 3 |
| 2 | 0–0.15 | 65 | 32 | 3 | 61 | 35 | 4 |
| 2 | 0.15–0.3 | 77 | 21 | 2 | 71 | 26 | 3 |
| 3 | 0–0.15 | 64 | 31 | 5 | 66 | 30 | 4 |
| 3 | 0.15–0.3 | 80 | 10 | 0.2 | 84 | 15 | 1 |
| 4 | 0–0.15 | 53 | 41 | 6 | 51 | 41 | 8 |
| 4 | 0.15–0.3 | 64 | 31 | 5 | 57 | 40 | 3 |

### 3.1.2. Root Distribution Modeling for Single Tree Systems

The RDM was calibrated and validated with the collected in-situ data. Two types of data are required to calibrate the model; (i) the distribution of fine roots in relation to the distance from the stem, and (ii) the frequency of various diameter classes at different distances from the stem. The best-fitted parameters of the root distribution model for "Tasman" poplar are presented in Table 4. $R^2$ of two dataset were 0.78 and 0.85, suggesting the model fitted quite well with the measured data (Table 5). We repeatedly fitted the parameters of the model randomizing the splitting of the training and testing datasets with the proportion of 80/20 to evaluate the model stability; the results are presented in sector Appendix A.

**Table 4.** Calibrated parameters of the root distribution model.

| Symbol | Parameter | Value |
|---|---|---|
| $\mu$ | Pipe coefficient | 97056.03 |
| $\beta$ | Empirical exponent of coarse root density | −1.501547 |
| $\eta$ | Scaling coefficient for maximum root diameter at a distance | 0.1319465 |
| $\psi$ | Proportionality constant for maximum root lateral extension | 16.21262 |

**Table 5.** Summary table of the calibration and validation of the root distribution model. 80% of total measured data (n = 140) was applied to calibrate the model whereas 20% (n = 32) was used to validate the model. SSE is the sum of square errors, and $R^2$ is the coefficient of determination.

| Dataset | n | SSE | $R^2$ |
|---|---|---|---|
| Training | 140 | 634.60 | 0.79 |
| Testing | 32 | 173.45 | 0.75 |
| Trench | 128 | 684.65 | 0.78 |
| Pit | 44 | 123.40 | 0.85 |

Figure 6 shows the fine root distribution of mean measured data and best-fitted modeled data considering different sizes of trees and various distances from the tree stems. The model underestimates fine root density in the first three trenches but well simulates it well in the farthest trench. The percentage errors between measured and modeled fine root density in the 1.5 m trench of the smallest to the biggest trees are 41%, 42%, 25%, and 10%. Similarly, the percentage errors between measured and simulated fine root density in the

farthest trench are 33%, 23%, 17%, and 36%. Overall, our findings indicate that the root distribution model tends to underestimate the number of fine roots in all trees.

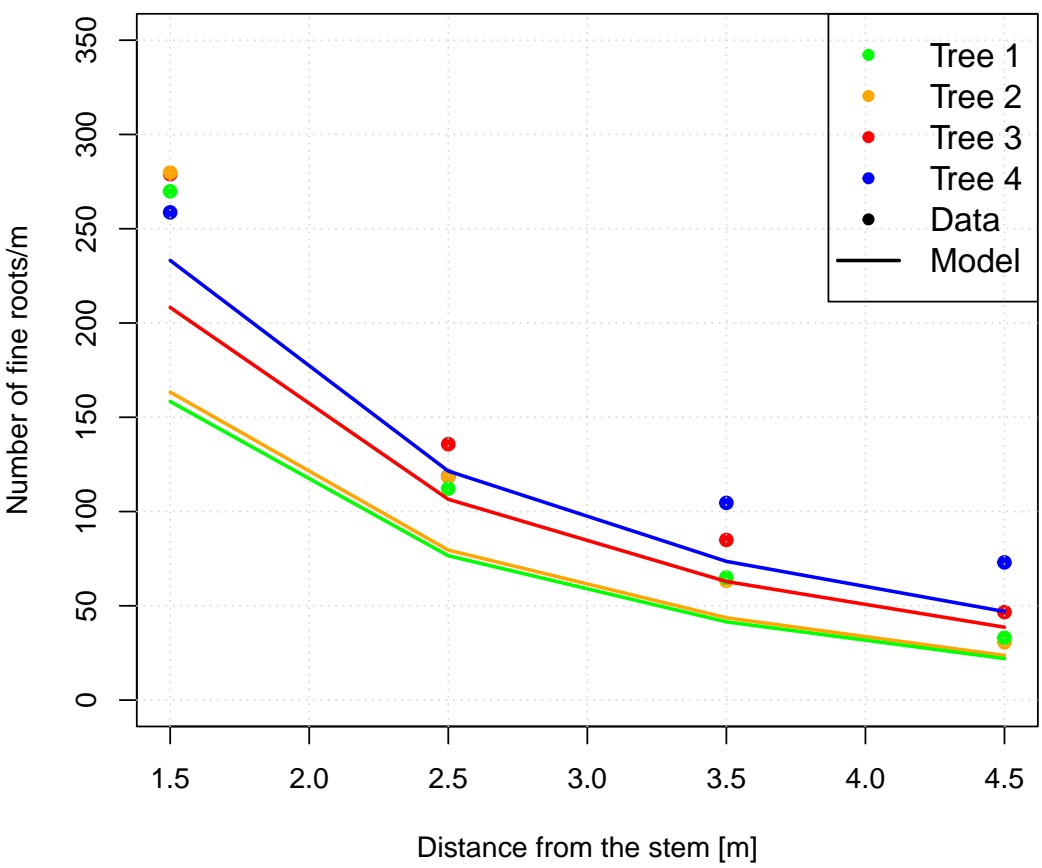

**Figure 6.** Mean measured (dots) and modeled (lines) number of fine roots per linear meter along the trenches (number/m), from four distances of 1.5 m, 2.5 m, 3.5 m, and 4.5 m from the four stems of "Tasman" poplars with different DBH. Tree 1 with 0.41 m DBH was presented in green, Tree 2 with 0.42 m DBH was indicated in orange, Tree 3 with 0.51 m DBH was exhibited in red, and Tree 4 with 0.56 m DBH was presented in blue.

The measured and modeled number of coarse roots from 1.5 to 10.5 mm diameter classes at each distance from the stem is presented in Figure 7. In general, the model tends to underestimate the abundance of coarse roots in all trenches of trees except from tree 3.

For the 10 mm class of root diameters, the differences between measured and predicted values in the 1.5 m trench from the smallest tree to the biggest tree are correspondingly 0.05, 0.13, 0.57, and −1.44 roots/m with simulated values higher than measured ones except from tree 4; whereas in 4.5 m trench, the variations are −0.17, −0.19, 0.11, and −0.47 roots/m, respectively with simulated values smaller than collected data in tree 1, 2 and 4. Therefore, the model exhibits a better estimation of coarse roots than fine roots. Generally, root distribution simulates the root density at each distance well with $R^2 = 0.78$.

### 3.1.3. Root Distribution Modeling at the Stand Scale

With the aim of validating our root distribution within a stand, we compared the simulated fine root abundance with measured data from four faces of the eleven soil pits in the transect. According to Figure 8, RDM performs well in the sparse zone in the stand (pits 1 to 5), whereas it tends to overestimate the number of fine roots in the pits situated in the denser zone (pits 6 to 11). In soil pit 1 and 2, which were located in the sparsest zone, the model estimates the number of fine roots with differences of 0.88 and 12 fine roots/m. In contrast, in pit 11, which was located in the densest zone, the modeled value

is around 2.6 times greater than the measured one, reaching up to 204 fine roots/m while mean measured value is just ca. 79 fine roots/m. The percentage errors between the mean measured and modeled fine root density in each pit from sparse to dense zone are 1%, 17%, 94%, 0.1%, 17%, 45%, 42%, 87%, 175%, 83%, and 159%.

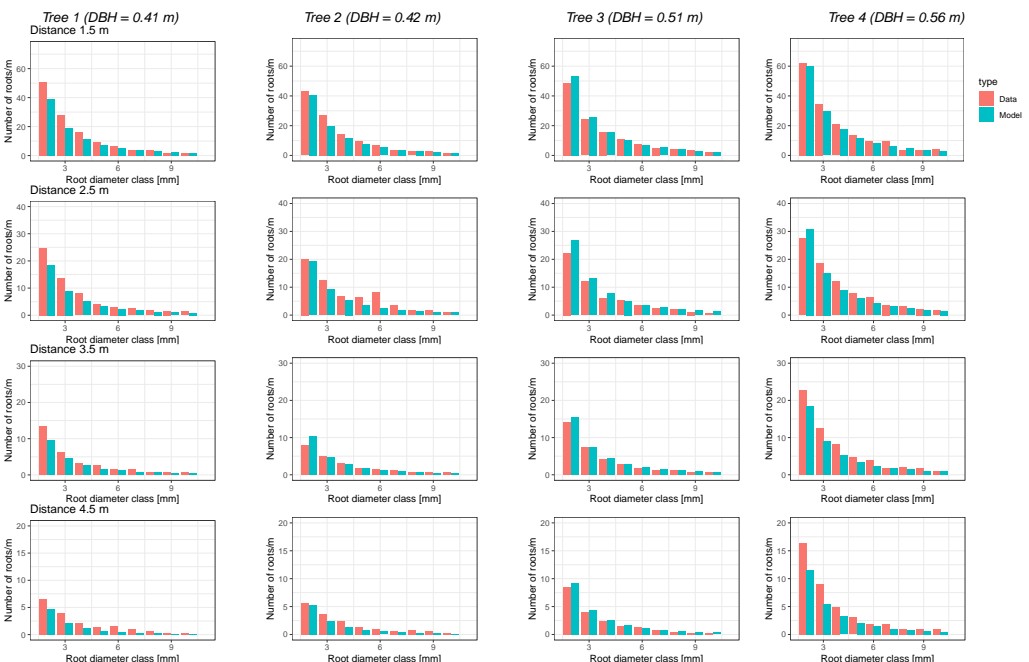

**Figure 7.** Measured (red bars) and modeled (blue bars) number of coarse roots in four poplar trees with different sizes at four distances of 1.5 m, 2.5 m, 3.5 m, and 4.5 m away from the stems. The red bars presented measured data whereas blue bars indicated simulated data.

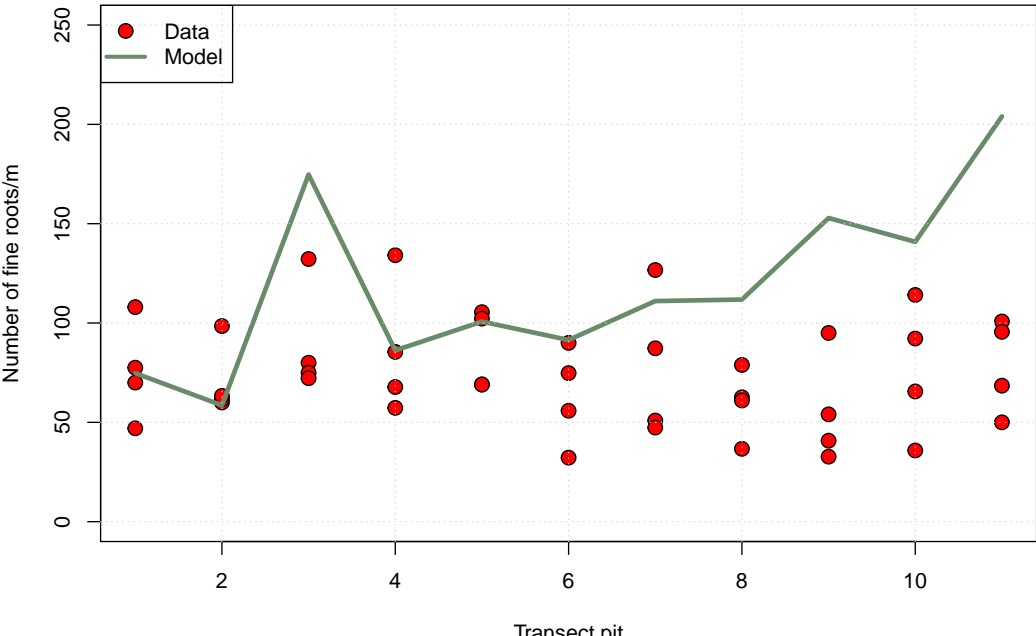

**Figure 8.** Comparison between measured and modelled number of fine roots in different transect pits (mean over the four soil faces). Red dots represent collected data at four soil faces of each pit; the green curve indicated the simulated average number of fine roots in 1 m width and 0.9 m depth of each pit.

Figure 9 compares the measured and modeled number of coarse roots in the transect of soil pits. Overall, the model overestimates the number of coarse roots, especially the 2 mm root class. The maximum and minimum variations between two types of data of the 2 mm root diameter class are recorded up to 32 roots in pit 11 and 0.72 roots in pit 1, respectively. The percentage errors of the predicted coarse root density compared to measured ones increase correspondingly from sparse to highly planting zone. We used the *t*-test to compare the means of two datasets and observed that the difference in number of roots in pit 1 between measured data (mean = 3.89) and simulated data (mean = 3.95) was insignificant (t(58) = 0.017112; *p* = 0.9864). The *t*-test performance on the comparison of measured and modeled root abundance in pit 11 showed that the number of roots simulated (mean = 10.80) was higher than measured values (mean = 4.06) but insignificant with a difference of (t(38) = 0.91348; *p* = 0.3669). Overall, the root distribution model performs well at the stand scale with $R^2$ = 0.85.

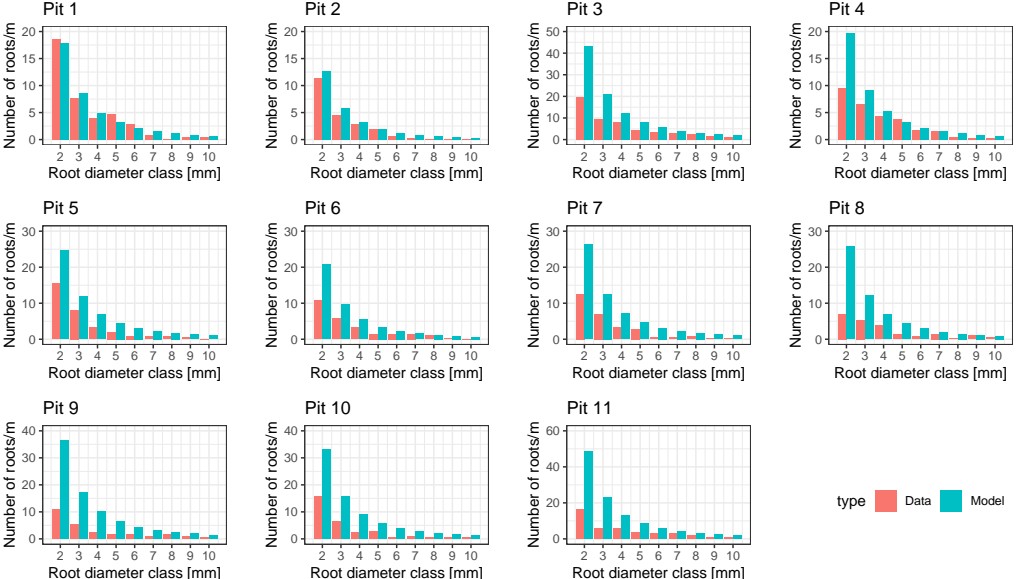

**Figure 9.** Comparison between measured and modeled number of coarse roots in root diameter class from 2 mm to 10 mm along the transect of soil pits in the stand (mean over the four vertical profiles of each pit).

### 3.2. Root Area Ratio

We calculated the cross-sectional area of roots per area of a soil profile (the root-area-ratio, RAR) at different distances from the stems.

Generally, most of the RAR is concentrated in the first 0.4 m soil depth and then decreases sharply close to 0 in all tree sizes, in accordance with the data on root distribution (Figure 10).

Results of RAR exhibit great variation with different tree sizes and distances to the stem (Figure 11). Generally, RAR values decrease when the distance from the stem increases. In all single tree datasets, the maximum RAR values are recorded in the closest trench (1.5 m) ranging from 1.048% to 0.69%. For the 1.5 m trenches, the maximum RAR value is measured in Tree 3 with DBH of 0.51 m, and the minimum value is recorded in Tree 1 with DBH of 0.41 m. The biggest differences between measured and simulated RAR values are found to be at the first trench, especially in tree 2 with a residual up to 0.43%. At the distance of 4.5 m, the differences between the two datasets are not large with values of 0.08%, 0.05%, 0.02%, and 0.07% corresponding to the tree sizes from smallest to largest. Overall, the root distribution model tends to underestimate the RAR of all trees at all distances. However, the model performs better at 3.5 and 4.5 m distance from the stems.

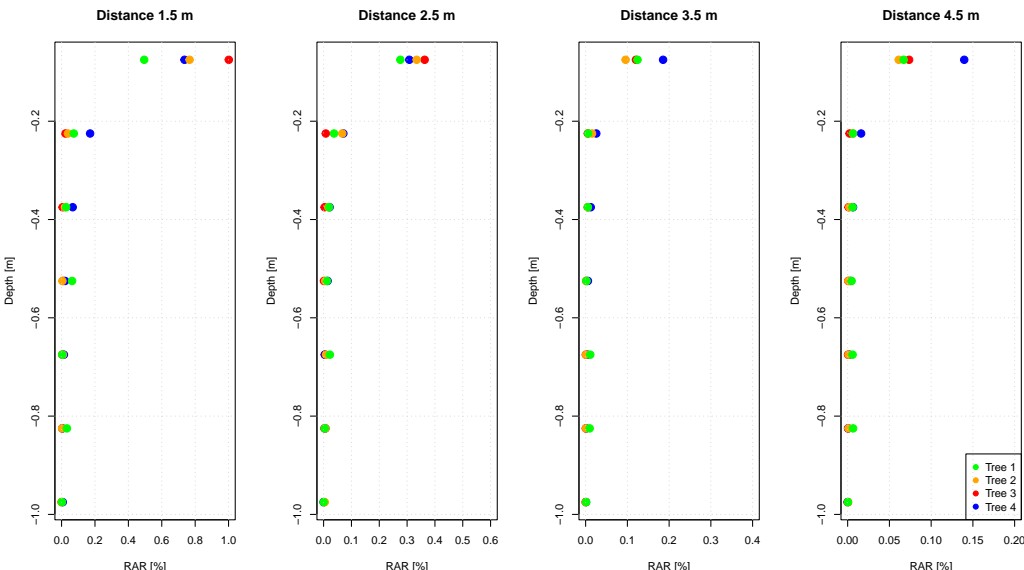

**Figure 10.** Correlations of measured RAR of four "Tasman" poplars with soil depth and distances from the stems. RAR values of tree 1 (DBH = 0.41 m) is represented in green dots, values of tree 2 (DBH = 0.42 m) is recorded in orange dots, values of tree 3 (DBH = 0.51 m) is indicated in red dots, and values of tree 4 (DBH = 0.56 m) is exhibited in blue dots.

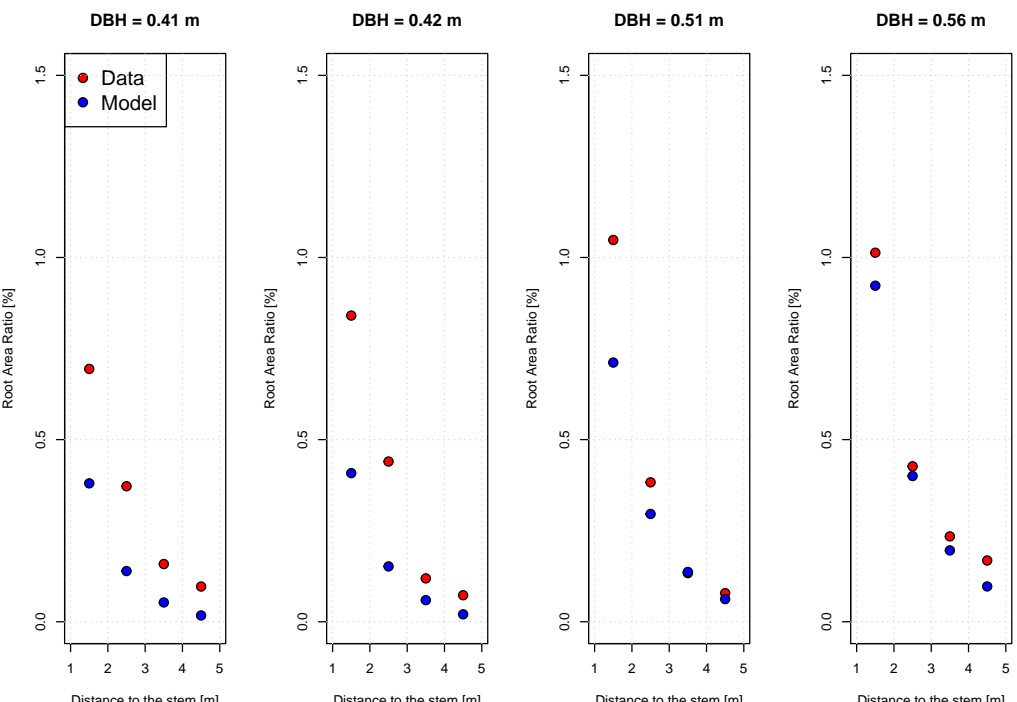

**Figure 11.** Comparison between measured (red dots) and simulated (blue dots) root-area-ratio (RAR) from different sizes of poplar trees at various trenches from the stems.

Along the transect of soil profiles, the biggest differences between collected and simulated RAR values are observed in transect pit 3, which is similar to the results of both fine and coarse root distribution (Figures 8, 9 and 12). The modeled RAR values are greater than the measured ones, even in pits located in the sparse zone. In contrast, the difference in RAR values between collected data and modeled is much smaller compared to the difference in fine root density in pit 11.

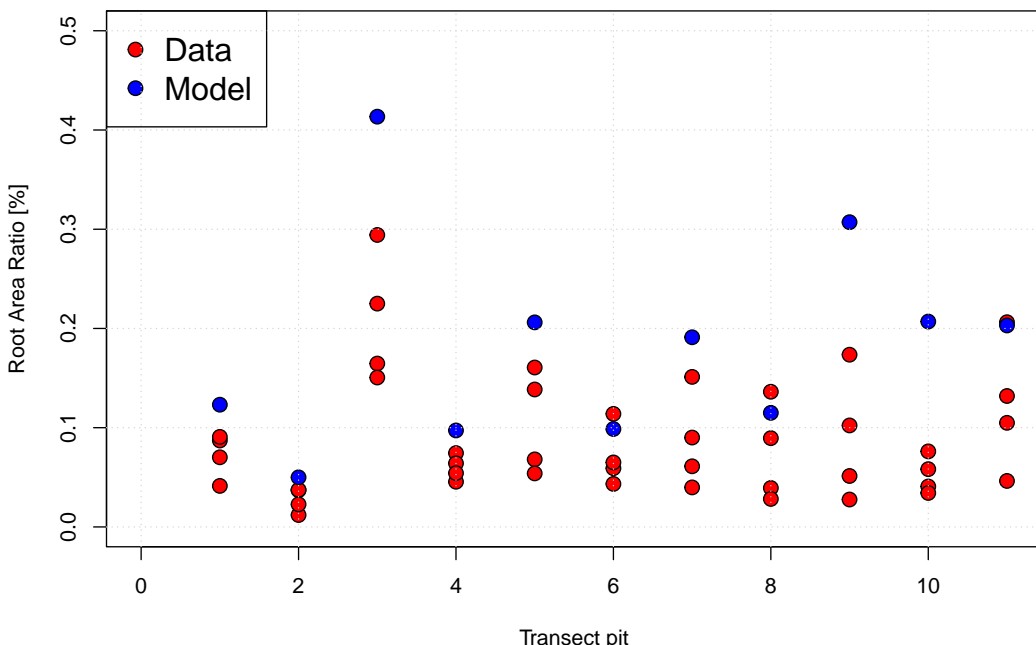

**Figure 12.** Comparison between measured (red dots) and simulated (blue dots) root-area-ratio (RAR) in various faces of transect's pits.

### 3.3. Root Reinforcement Upscaling

#### 3.3.1. Root Bundle Model

The mechanical properties of poplar's roots were quantified using both laboratory tensile tests (124 tests) and pullout tests (66 tests) with root diameters ranging from 1.7 mm to 40 mm. Figure 13 shows all 190 collected data and presents a clear increase in maximum tensile force with increasing diameter. A shape increase in data variability can be observed in roots from 0.005 m to 0.01 m in diameter. However, with roots from 0.01 m and thicker, the rise appears to remain rather constant. The fitting correction with the cumulative normal distribution function (power-law fit + Survival curve) visibly diminishes residuals for small roots, especially roots smaller than 0.01 m. The $R^2 = 0.88$ suggests that the curve predicts the measured tensile forces well.

Figure 14 shows the survival function estimated from both root tensile tests in laboratory and pullout tests in-situ. The best-fitted Weilbull shape factor $\omega$ was 1.83 and the scaling factor $\lambda$ was 1.53.

Table 6 summarizes the calibrated parameters of the root mechanical properties required for the root reinforcement quantification with the RBMw model of "Tasman" poplars.

**Table 6.** Calibrated parameters of the RBMw model.

| Symbol | Parameter | Value |
|---|---|---|
| $F_0$ | Root force scaling factor | $2.9 \times 10^6$ |
| $\alpha$ | Root force shape factor | 1.55 |
| $k_0$ | Root spring constant scaling factor | $9.7 \times 10^6$ |
| $\lambda$ | Weibull scaling factor | 1.53 |
| $\omega$ | Weibull shape factor | 1.83 |
| $\phi_m$ | Mean of cumulative normal distribution | 0.00643 |
| $\phi_{sd}$ | Standard deviation of cumulative normal distribution | 0.00365 |

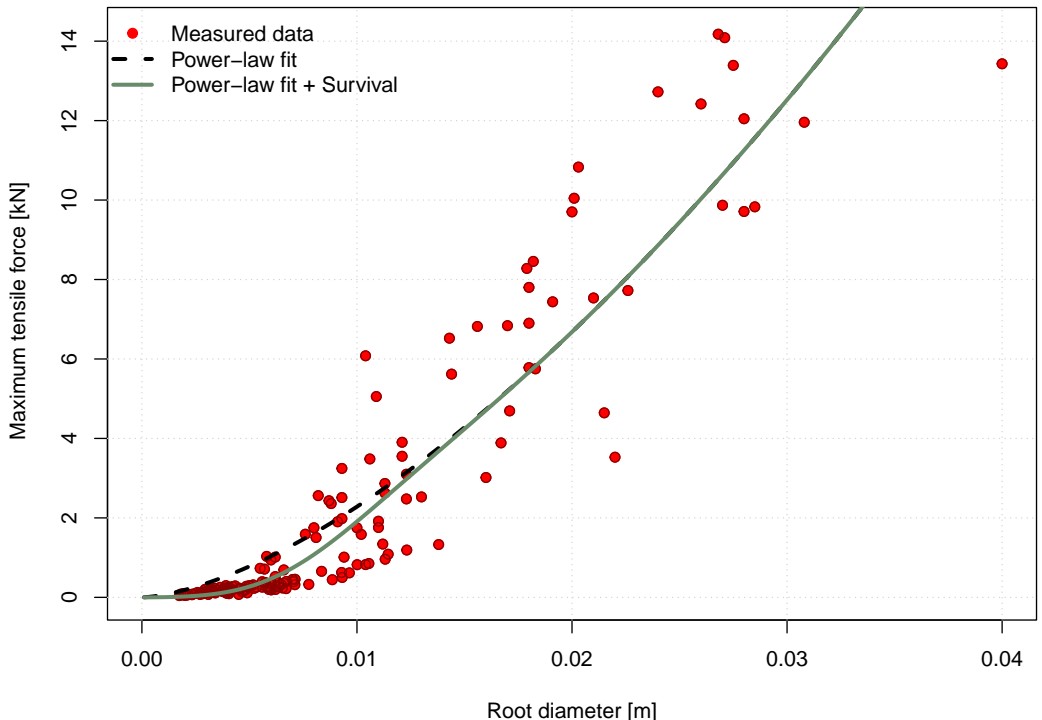

**Figure 13.** Maximum tensile force in relation to root diameter of "Tasman" poplars.

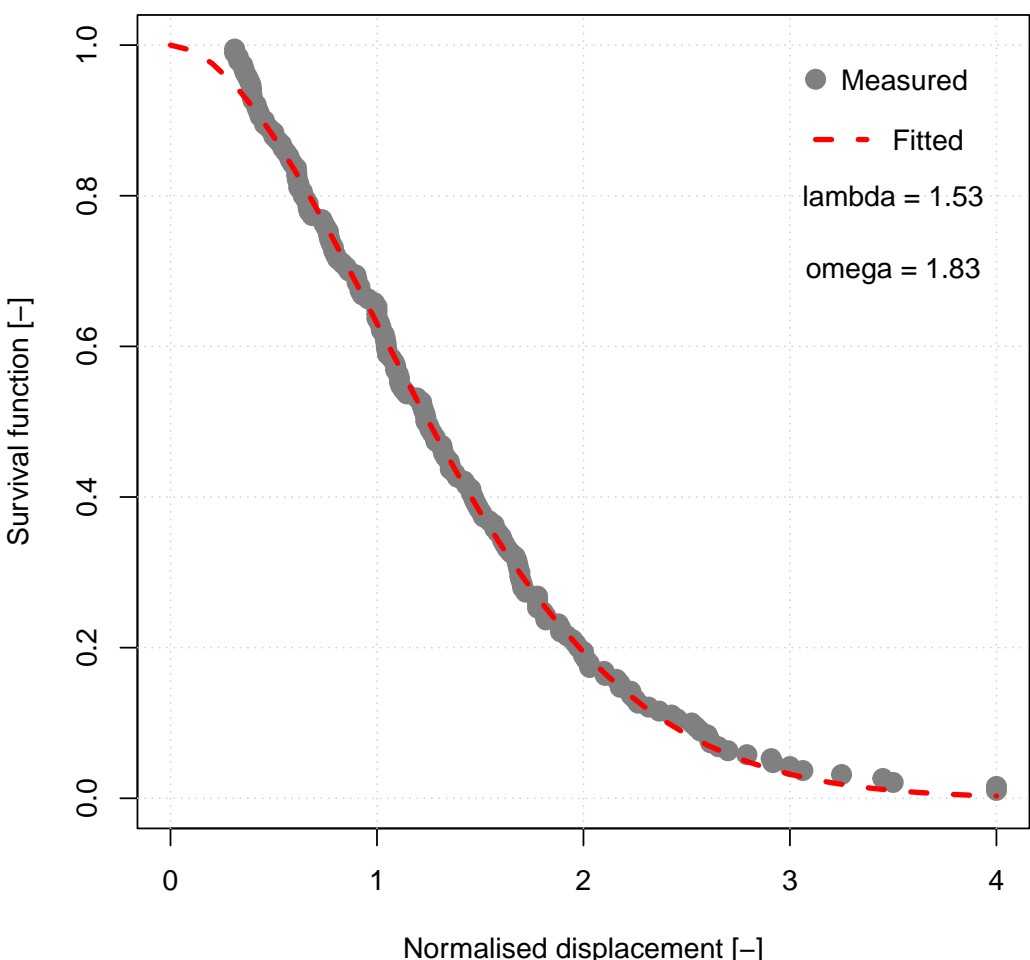

**Figure 14.** Survival function for collected data from both laboratory tensile test and pullout test. Grey dots are measurements. The red dashed line represents the best fit.

### 3.3.2. Root Reinforcement as a Function of Distance and DBH

In general, lateral root reinforcement increases with increasing stem sizes and decreases with increasing distances from the stem. The coefficients fitted by the minimum Sum of Squared Errors in Equation (12) are listed in Tables 7 and 8. $R^2$ of the training and testing dataset are 0.60 and 0.69, respectively, indicating that the model performs reasonably well. Figure 15 compares measured (dots) and simulated (lines) lateral root reinforcement as a function of stem size (DBH) and distances from the stem.

**Table 7.** Calibrated parameters of the root reinforcement model.

| Symbol | Parameter | Value |
|--------|-----------|-------|
| a | Scaling factor | 41030.49 |
| b | Shape parameter | 0.9892003 |
| c | Rate parameter | 9.750829 |

**Table 8.** Summary table of the calibration and validation of the root reinforcement model. 80% of total measured data (n = 140) was applied to calibrate the model whereas 20% (n = 32) was used to validate the model. SSE is the sum of square errors, and $R^2$ is the coefficient of determination.

| Dataset | n | SSE | $R^2$ |
|---------|---|-----|-------|
| Training | 140 | 1157700 | 0.60 |
| Testing | 32 | 275801 | 0.69 |
| Trench | 128 | 1259770 | 0.64 |
| Pit | 44 | 173731 | 0.32 |

The results of the model show that a single tree with 0.6 m DBH is able to contribute to soil strength with a root reinforcement up to 64 kN/m in the 1.5 m trench and still provide up to 4.5 kN/m reinforcement at 4.5 m distance. However, a single tree with 0.4 m DBH can reach ca. 22 kN/m at 1.5 m distance and around 0.4 kN/m at 4.5 m distance from stem. Based on Figure 15, there seems to be insignificant influence of DBH on the root reinforcement, especially from 2.5 m distance and upwards.

Figure 16 shows the residuals of the modeled lateral root reinforcement as a function of distance from the tree stem and the DBH. The data shows a higher variability of the residual near the tree stem and a decrease with increasing distance from the stem. The variability of residuals is similar between the DBH values. The greatest variations are up to 40 kN/m in the 1.5 m trench and 15 kN/m in the 4.5 m trench. The proportion of the positive values of residuals is higher than the negative ones, especially at greater distances from the tree stem. However, the mean values of variances from the closest to the farthest trenches were 9.4, 9.4, 4.03, 4.15 kN/m respectively, indicating a general underestimation of the model.

Along the transect, the modeled lateral root reinforcement values tend to be in good agreement with the measured ones. Only for the trenches with high tree density, the model tends to overestimate (Figure 17). Interestingly, the highest and lowest measured forces are in two neighbor pits in the low density part of the stand. The highest value is in the pit 3 (by the nearest distance between 2 trees), whereas the weakest one is in the pit 2 (by the farthest distances between 4 trees).

### 3.3.3. Vertical Distribution of Root Reinforcement

Most lateral root reinforcement is concentrated within the first 0.30 m of soil depth, in both the trenches and transect pits. Equation (13) was calibrated with the measured data, shown with the red points and lines in Figure 18. The best-fitted values of the parameters are summarised in Tables 9 and 10 with an overall $R^2$ value of 0.99.

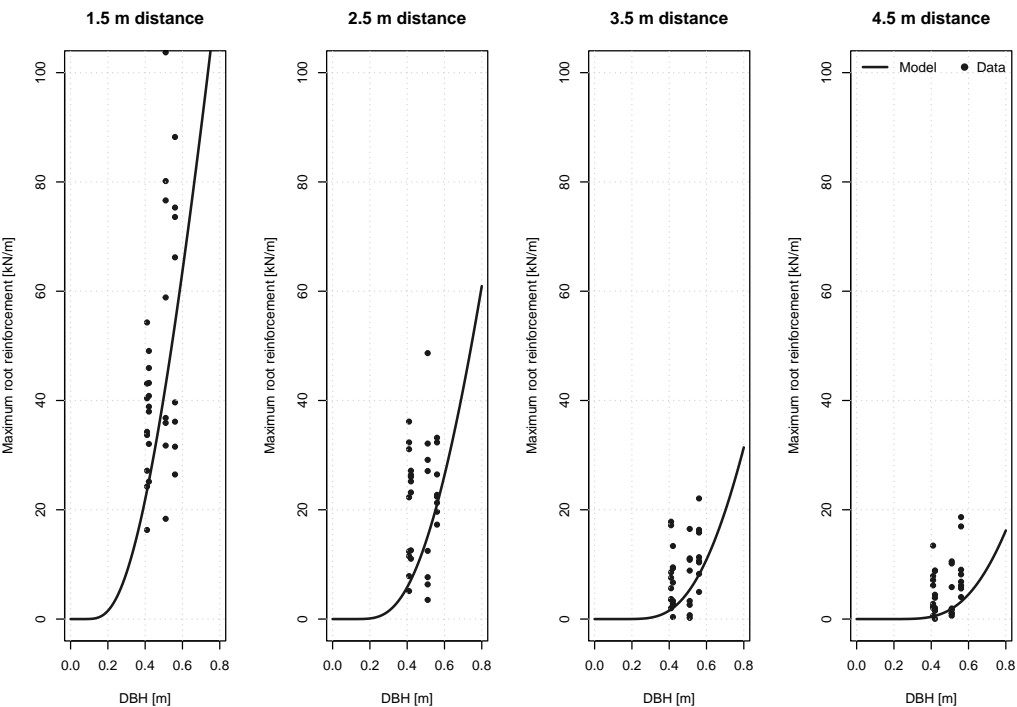

**Figure 15.** Maximum root reinforcement as a function of the tree DBH at four distances from the stem. Black dots represent measured root reinforcement calculated with RBMw while the black line indicates root reinforcement estimated with maximum lateral root reinforcement.

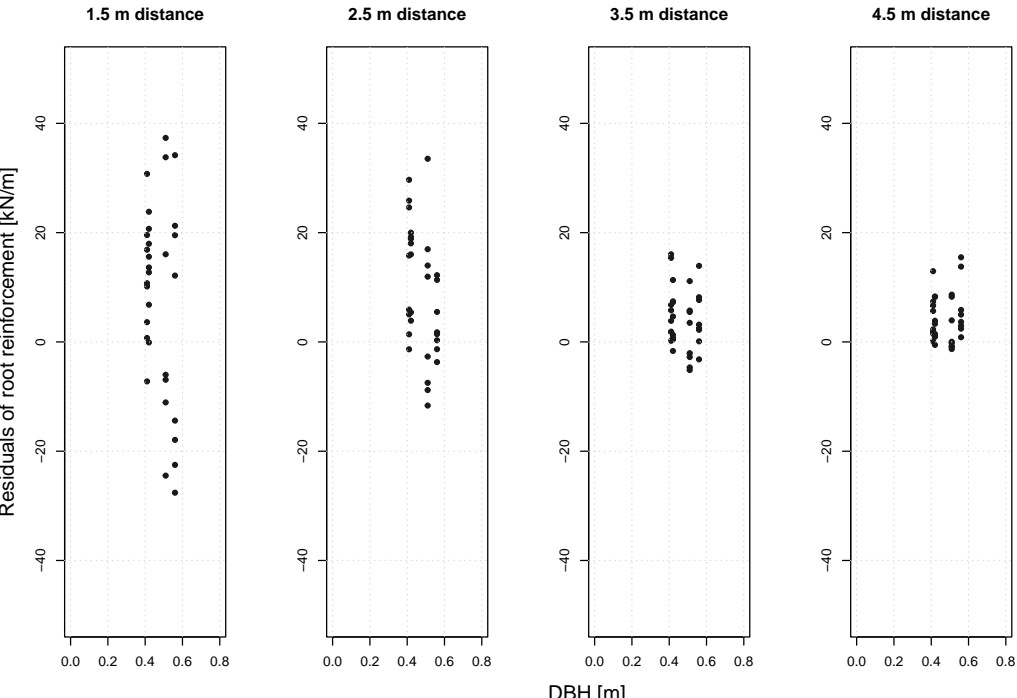

**Figure 16.** Residuals of the modeled lateral root reinforcement.

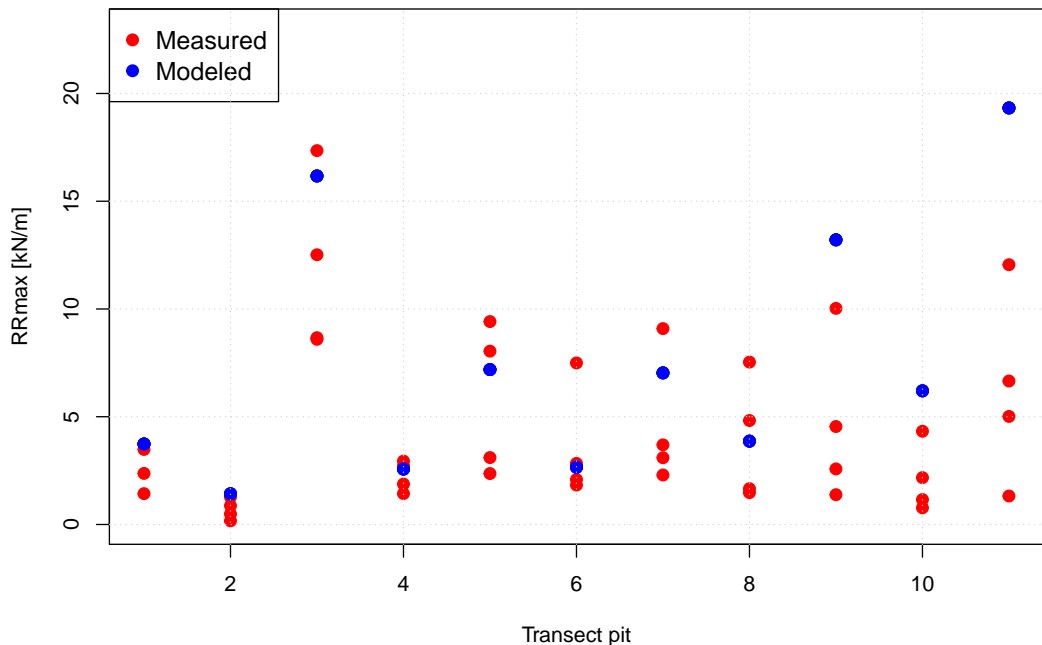

**Figure 17.** Comparison of the modeled (blue dots) and measures-based (red dots) values of root reinforcement along the transect of pits in the poplar stand. The modeled values are calculated for the center of the pits, whereas the measures-based values are calculated for each of the four profiles of the pits.

**Table 9.** Calibrated parameters of the root reinforcement model.

| Symbol | Parameter | Value |
|--------|-----------|-------|
| $Z_\alpha$ | Shape parameter | 1.151732 |
| $Z_\beta$ | Rate parameter | 14.98385 |

Calibration and validation parameters of the model for the vertical distribution of root reinforcement (n = 172) with $Z_\alpha$ is the shape parameter, $Z_\beta$ is the rate parameter, and SSE is the sum of square errors.

**Table 10.** Summary table of the calibration and validation of the root reinforcement model. 80% of total measured data (n = 140) was applied to calibrate the model whereas 20% (n = 32) was used to validate the model. SSE is the Sum of Square Errors, and $R^2$ is the coefficient of determination.

| Dataset | n | SSE | $R^2$ |
|---------|---|-----|-------|
| Training | 140 | 0.09 | 0.99 |
| Testing | 32 | 0.15 | 0.99 |
| Trench | 128 | 0.16 | 0.99 |
| Pit | 44 | 0.09 | 0.99 |

Within the first soil layer, the maximum root reinforcement value is measured up to 103.72 kN/m in the 1.5 m trench of Tree 3. The mean root reinforcement in each soil depth among the soil trenches are 17.12, 1.90, 0.80, 0.50, 0.34, and 0.30 kN/m, respectively. The strongest root reinforcement among transect pits is found to be 16.67 kN/m located in the first soil layer of pit 3. From 0.3 m and downwards, the basal root reinforcement decreases very rapidly. Among the second soil depths, the maximum root force is recorded in the first trench of Tree 4 with a value of 36 kN/m and in pit 11 with a value of 3.17 kN/m. At the average depth of 0.375 m, the strongest force is found in the first trench of the biggest tree—Tree 4 with values of 10.48 kN/m while at the depth of 0.525 m, the maximum force is recorded in the first trench of Tree 1 with 16.44 kN/m. From 0.6 m and downwards,

the basal root reinforcement of all soil profiles is so weak that its contribution to soil strength would be insufficient. Unfortunately, we could not collect data from the roots under the stem. Vertical root distribution was recorded at 1.5, 2.5, 3.5, and 4.5 m from the stem. However, one pit was found to have a thick root of up to 100 mm running vertically. Such a root would have great influence on the basal root reinforcement, explaining the big variation of root reinforcement at the soil layer 0.8 m in pit data. According to Figure 18, most of the basal root reinforcement concentrates in the first soil layers and decreases sharply following deeper layers.

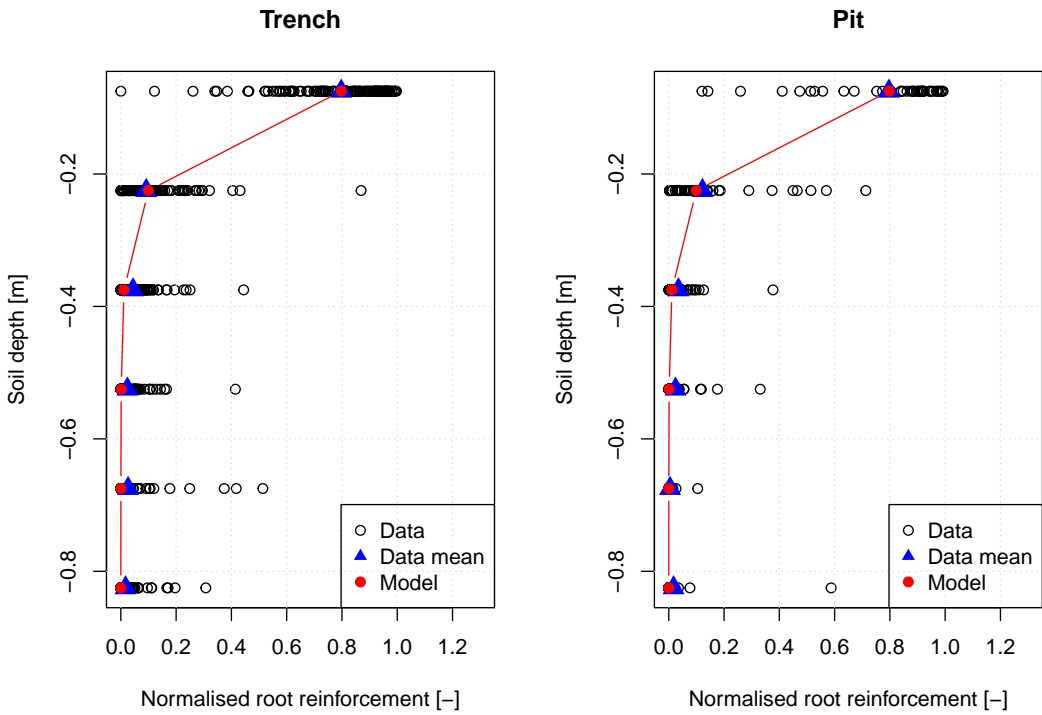

**Figure 18.** Normalised basal root reinforcement as a function of soil depth. Blank dots represent measured data, blue triangles represent mean measured normalised root basal reinforcement, and red dots show the modeled data.

## 4. Discussion

### 4.1. Distribution of the RAR

According to Gasser et al. [27], the root-area-ratio is one of the key factors to quantify root mechanical and hydrological effects on soil strength on hydraulic bank erosion. In this study, the values of RAR increase with increasing DBH and decrease with increasing soil depth and distance from the tree stem. These results are similar to others reported in the literature [14,44,68,69]. However, the values of RAR measured in this study are two orders of magnitude higher than the values reported for the same stand at the age of 9 years old. The DBH reported in Douglas et al. [70] ranged between 0.173 and 0.192 m for tree densities of 84 and 770 sph, respectively. The values of RAR reported in Douglas et al. [70] ranged between 0.00001 and 0.00003% at a distance of 0.9–1.8 m. These values are surprisingly low compared to the values reported by Zydroń et al. [71] for black poplars with RAR up to 0.44% and on average 0.225% for an 8-year-old tree at 0.5 m distance, grown in Poland. The values of RAR calculated in this study at age 26 years, in the same study area as Douglas et al. [70], reach values that ranged between 0.02 to 1% at a distance of 1.5 m from trees with a DBH of 0.41-0.56 m (tree density of ca. 84 sph). This huge increment in RAR within 17 years reflects the rapid growth potential of poplar root systems.

The majority of the RAR is located in the first 0.4 m soil depth, confirming the results of similar studies on younger stands [70,71]. In the specific case of the Ballantrae study area, the limitation in the vertical distribution of roots is mostly due to the type of soil, where the

hydromorphological characteristics of the mineral horizons indicate a clear influence of water fluctuation. Additionally, other factors may contribute to confine the root distribution in the upper soil horizons, as discussed in Douglas et al. [70]. Nevertheless, some roots were measured at depths greater than 0.8 m too, as confirmed in Douglas et al. [70] and Zydroń et al. [71] for younger trees.

The measured RAR values among different soil faces of the transect pits suggest that the root distribution can be greatly influenced by the distance to the nearest tree and its DBH more than the characteristics at stand scale. Roots with diameters ranging from 4–10 mm contribute the highest to the value of the RAR, as shown in other studies [69]. In comparison with pit 3 situated where tree density is lower (84 sph), pit 10 was located in an area of greater tree density (770 sph) and has much lower RAR values. This is probably due to the distance between the nearest tree and its DBH. This strong influence of the nearest tree is also confirmed by the model results. As observed from Table 2, pit 3 was 3.2 m far from a tree of 0.55 m DBH, whereas pit 10 was 4.5 m from a tree with a 0.47 m DBH. Moreover, pit 3 had a higher root density than pit 7, located at a similar distance to the nearest tree, but the tree size was smaller with 0.46 m in DBH. Similarly, pit 1 also shows a slightly higher value in RAR compared to pit 10 due to the DBH of the nearest tree. Other studies have shown that for the high planting density of younger trees, the influence of the DBH on root distribution is less dominant than in older plantations [72]. Moreover, several studies have shown that measured root density distributions are affected by several factors such as tree species, climate, sampling time/season, soil type, land use management, and orientation of soil trenches [10,70,73–77]. The differences in DBH values measured within the analysed 26-year-old poplar stand are clearly correlated with the stand density and thus with the competition for resources (light and nutrients). The initial size of the poles used for the plantation may also have had an influence, as discussed in Phillips et al. [8] and Schwarz et al. [33]. As observed in Table 2, the average DBH values of all trees decrease with increasing planting density, from 0.58 m in pit 1 to 0.47 m in pit 11. The results of the root distribution show that considering the influence of the tree dimension using the "pipe-theory" [12,78] is an appropriate assumption for a single tree, less influenced by the concurrence of neighboring trees (such as in spaced planted tree conditions). However, the model seems to produce poorer predictions in the case of densely planted trees, due to stronger competition between neighbors. Future research needs to focus on how the influence of tree-neighbor competition in densely spaced trees induces a lateral and vertical optimisation of the roots occupancy, drifting the shape of the root system from a symmetrical-circular-like shape to an irregular one, defined by the position of the neighbor trees. This effect was previously discussed in Phillips et al. [8], where root growth in densely planted poplar tends to occupy unplanted areas.

*4.2. Spatial and Temporal Distribution of the Root Reinforcement, and Its Implication for Shallow Landslide Stabilisation*

Basal and lateral root reinforcement are the principal mechanisms that contribute long-term to the prevention of shallow landslides [19]. The results of this study show that basal root reinforcement is limited to the first 0.4 m depth, with low values extending deeper than 0.8 m (mostly near the tree stem), whereas lateral root reinforcement reaches values up to 20 kN/m at a 4.5 m distance from single isolated trees. Within the stand, the values of lateral root reinforcement are strongly influenced by the distance from tree stems and their dimensions, analogous to the observation made for the distribution of RAR. However, the model tends to overestimate root reinforcement for tree densities higher than 200 sph. Poplars would not normally be planted at densities higher than this in practice. Considering that for wide-spaced tree planting measures to control erosion, the tree density is usually less than 200 sph, the validated model can be applied for that condition. As discussed in Schwarz et al. [33], the optimum stand density for erosion control, carbon sequestration, and pasture productivity corresponds to a tree canopy cover of 30%, which corresponds to about 70 sph for a mean DBH of 0.3 m and to about 30 sph

for a mean DBH of 0.53 m. This would lead to values of lateral root reinforcement near 0. In order to ensure sufficient root reinforcement, stem densities between 160 and 330 sph are needed, confirming the indication given in Schwarz et al. [33]. Based on the result of this study, this range of stand density would guarantee a lateral root reinforcement between 1 and 16 kN/m for a mean DBH of 0.5 m in a triangular lattice of trees, following the approach described by Flepp et al. [22].

The vertical distribution of root reinforcement determines the amount of basal root reinforcement. As previously discussed, the site conditions limit most of the roots on the first 0.4 m soil depth. It is documented that basal root reinforcement is significantly affected by the characteristics of the studied site and the depth of the potential shear plane of a landslide [55,79]. However, even a low value of basal root reinforcement at 0.5–1.0 m of perpendicular depth, corresponding to the thickness of most shallow landslides in NZ, may have a major contribution to slope stability. Especially considering that in correspondence with each tree stem, sinker roots as observed by McIvor et al. [47] act locally as anchors transferring forces of the superficial root networks deeper in stable soil layers.

The temporal variation of root reinforcement may be mainly due to two types of processes: one is due to the dynamic of root distribution and mechanical properties during different seasons, and the second is due to the tree growth and stand dynamic over the years. McIvor et al. [13] found that "Tasman" poplar has a significant reduction in fine-root length density during the dormant season, whereas coarse root distribution shows little change. Considering that, due to the fact that coarse roots dominate the contribution of root reinforcement [21,33], no significant seasonal changes are expected in root reinforcement where coarse roots are present. Little is known about the seasonal changes in the mechanical properties of poplar roots and more needs to be explored in future research, as discussed in Schwarz et al. [33]. However, the fitting of the force-root diameter model shows a good agreement between the two different datasets. A clear positive correlation between maximum tensile force and root diameter is observed, similar to previous findings of poplars [71,80], indicating that even in different growing conditions, different sampling seasons the obtained values are quite similar.

Over the years, tree root systems develop, increasing their capability to effectively stabilise soil on steep slopes. McIvor et al. [47] concluded that poplar trees, which were situated on erosion-prone slopes, needed at least 5 years to obtain a structural root network sufficient to stabilise soil. The poplars analysed in the Ballantrae study site over the years [33,45], showed a considerable constant DBH-growth rate of about 0.019 m/year (slightly influenced by stand density). Under this condition, a stand with a density of 100 sph (about 10 m distance between trees), would reach a minimum lateral root reinforcement in between the four or three neighbor trees of about 1.1–1.9 kN/m only after 30 years; whereas for a stand density of 250 sph (about 6 m distance between trees) after 30 years, the minimum lateral root reinforcement would reach values larger than 13 kN/m. An overview of the estimated minimum lateral root reinforcement as function of stand age is given in Table 11. These results are an important basis for the formulation of guidelines for the planning of erosion control measures using wide-spaced trees in New Zealand pasture hill country.

Compared to other tree species, the lateral root reinforcement of "Tasman" poplar results is greater within the first 1-2 m distance from stem than chestnut (*Castanea sativa*) [15] or spruce (*Picea abies*) [22]. For the same tree size of 0.5 m, "Tasman" poplars had the highest root force in the first 3.0 m and rapidly reduced to about 1 kN/m at a 5.0 m distance. Among the three species, chestnut trees have the highest values of root reinforcement at the largest distances from the stem (Figure 19).

**Table 11.** Calculated dynamic of lateral root reinforcement (kN/m) for different stand densities, based on the results of this study. The results are calculated for the minimum expected value within a stand with squared (lower values) or triangular lattice (higher values), following the approach described in van Zadelhoff et al. [25]

| Stand Density | Distance between Trees in a Squared Lattice | Root Reinf. 10 Years | Root Reinf. 20 Years | Root Reinf. 30 Years |
|---|---|---|---|---|
| sph | m | kN/m | kN/m | kN/m |
| 100 | 10.0 | 0 | 0–0.1 | 1.1–1.9 |
| 150 | 8.2 | 0 | 0.2–0.3 | 3.8–5.6 |
| 200 | 7.1 | 0 | 0.5–0.9 | 7.9–10.6 |
| 250 | 6.3 | 0 | 1.0–1.7 | 13.0–16.4 |

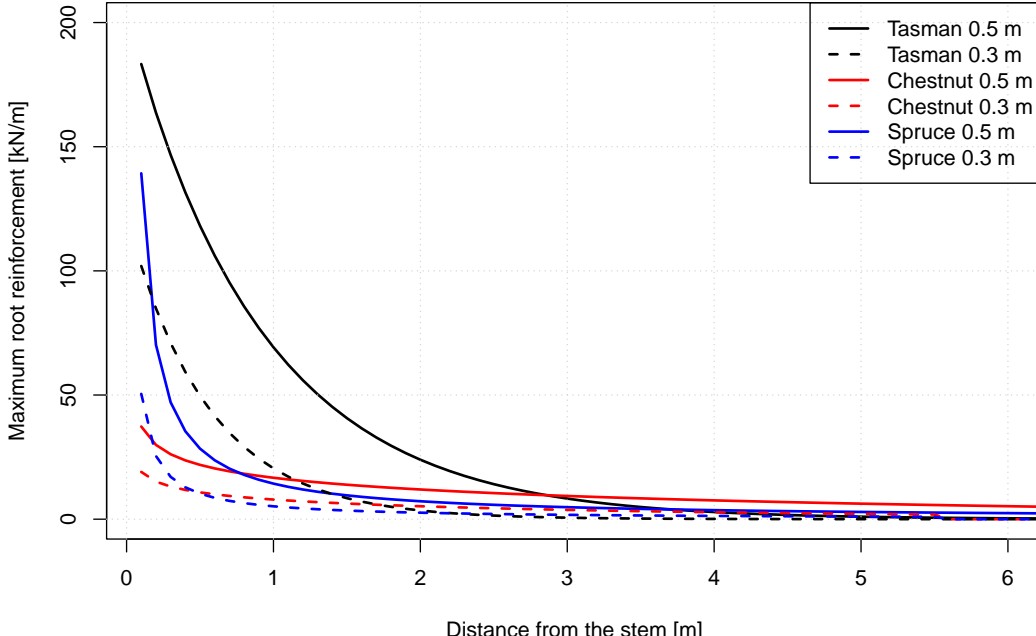

**Figure 19.** Compared maximum lateral root reinforcement of "Tasman" poplar with calibrated parameters from the present study, of chestnut coppices from the study of Dazio et al. [15], and spruce from the study of Flepp et al. [22].

## 5. Conclusions

This study provides a unique and detailed dataset of root distribution and mechanical properties of "Tasman" poplar growing in the pastoral hill country in New Zealand. Moreover, numerical models for root and root reinforcement distribution have been calibrated and validated for the first time using a combination of single tree excavations and a transect along a gradient of stand densities. Additionally, 124 laboratory tensile tests were combined with 66 in-situ pullout tests to quantify the root pullout forces for root diameters up to 0.04 m.

The collected data extends the research from previous studies allowing for the quantification of the temporal dynamics of root distribution and reinforcement over 26 years of tree growth, considering different stem densities. In general, the high growth rate of the young trees is also confirmed in older trees and is reflected in the increment of root distribution and reinforcement as a function of the DBH. The results show that at least 20 years are needed to reach a minimum value of lateral root reinforcement at the stand scale, and at least 30 years are needed to reach root reinforcement sufficient to stabilise most of the shallow landslides depending on their disposition, as discussed in Schwarz et al. [33].

The applied root distribution model well estimates spatial root distribution in individual poplar trees ($R^2$ = 0.78) and within a stand with low density, whereas it tends to overestimate the number of roots in the stand with stem densities higher than 200 sph. We suggest improving root distribution model performance in dense stands by adding a threshold into the model to limit overestimation.

The lateral root reinforcement model has a trend of underestimating root force in individual trees ($R^2$ = 0.64), whereas it performs well along the transect in the stand with tree stem densities lower than 200 sph. The model also predicts the vertical distribution of root reinforcement well, which is mostly limited to the first 0.4 m of soil depth.

The results presented in this paper allow the implementation of the temporal and spatial distribution of root reinforcement in numerical models for the estimation of the effectiveness of different types of bio-engineering measures to reduce soil erosion due to hydraulic bank erosion and shallow landslides [25,27,33]. Moreover, these tools are fundamental to develop strategies to prioritise interventions and optimise investments in green-based solutions at large spatial scales.

Further studies are needed to extend the application of the results, knowledge, and tools discussed in this paper for other plant species considering a wide range of environmental conditions, including the effects of climate change.

**Author Contributions:** Conceptualization, M.S.; methodology, M.S.; software, M.S.; validation, M.S. and H.M.N.; formal analysis, H.M.N.; investigation, F.B.v.Z., I.G., J.P., G.F., L.D. and C.P.; resources, M.S. and C.P.; data curation, H.M.N. and M.S.; writing—original draft preparation, H.M.N.; writing—review and editing, H.M.N., F.G. and M.S.; visualization, H.M.N., M.S. and F.B.v.Z.; supervision, M.S.; project administration, M.S. and C.P.; funding acquisition, M.S., C.P. and F.G. All authors have read and agreed to the published version of the manuscript.

**Funding:** This research was funded by the Ministry of Business, Innovation and Employment of New Zealand, the Bern University of Applied Sciences, and the University of Sassari.

**Data Availability Statement:** Information of The Emissions Trading Scheme (ETS) can be found at http://www.maf.govt.nz/environment-natural-resources/emissions-trading-scheme. Additionally, The New Zealand Poplar & Willow Research Trust (https://www.poplarandwillow.org.nz/) provides essential information about poplar and willow trees in pastoral hill country in NZ. Both RDM and RBMw model were implemented in R software and can be downloaded at the following link www.ecorisq.org/openFTP/Schwarz.zip. The detailed 3D visualization of the study site is available at https://fvzadelhoff.nl/Potree/nz_2/OurWebViewer.html, accessed date 1 March 2023.

**Acknowledgments:** The results are part of the STEC Project. Ha My Ngo is supported by a doctoral fellowship from the Department of Agriculture of University of Sassari and a mobility grant from the Bern University of Applied Sciences.

**Conflicts of Interest:** The authors declare no conflict of interest.

**Abbreviations**

The following abbreviations are used in this manuscript:

| | |
|---|---|
| ASL | Above sea level |
| DBH | Diameter at the breast height |
| kN | Kilonewton |
| NZ | New Zealand |
| RAR | Root-area-ratio |
| RBMw | Root Bundle Model with Weilbull survival function |
| RDM | Root Distribution Model |
| RR | Root reinforcement |
| SSE | Sum of Squares Error |
| sph | Stems per hectare |

**Appendix A**

Figure A1 displays boxplots of normalised SSE of 30 random generated combinations of training/testing datasets (80/20) for each proposed model. In Figure A1a, the difference in normalised SSE between training and testing datasets fluctuated from ca. $-187$ to 283 N of cumulative root-force. Whereas for the testing dataset, the variability of the normalised SSE was higher. For maximum lateral root reinforcement, the difference varied greatly from $-6 \times 10^2$ kN/m to $4 \times 10^2$ kN/m; however, the mean value of differences was $-1.242$ kN/m (Figure A1b). Lastly, the residuals between modeled and measured values of basal root reinforcement varied from $-253$ N to 256 N. Nevertheless, the mean value of the residuals was just 4.7 N. The mean normalised SSE of all models were quite close to 0, suggesting that in general, the models converged to similar accuracy in the training as well as in the testing results.

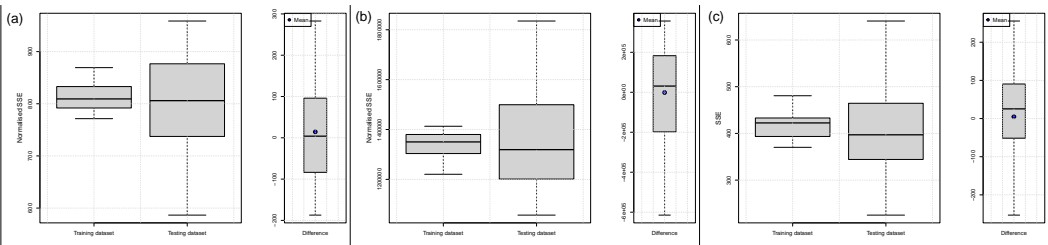

**Figure A1.** Normalised SSE of training dataset, testing dataset and residuals of (**a**) Root distribution model, (**b**) Lateral root reinforcement model, and (**c**) Basal root reinforcement model.

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
