# Peer review of "Analysis of Poplar’s (Populus nigra ita.) Root Systems for Quantifying Bio-Engineering Measures in New Zealand Pastoral Hill Country"

_forests, doi:10.3390/f14061240_

Round 1
Reviewer 1 Report (New Reviewer)
This study is very interesting, and could be published after revising.
The Introduction and Methodology of the manuscript is okay, but the results and discussion part need to be further revised and refined.
Total 23 figures and 11 tables are too much! Authors need to refine the manuscript to show the key points of root systems. In addition, more discuss about the data should be added.
Author Response
Dear Reviewer,
Firstly, thank you very much for your clear and kind comments on the manuscript. We greatly appreciate your time dealing with the manuscript and your support.
Our responses to your comments and modifications of the manuscript have been stated in the attached PDF file.
We appreciate all your constructive suggestions. We are in the process of submitting the revised manuscript based on your comments. Thank you very much.
Best regards,
On behalf of the co-authors

Reviewer 2 Report (New Reviewer)
Overview
The manuscript by Ha My Ngo, "Analysis of poplar’s (Populus nigra ita.) root systems for quantifying bio-engineering measures in New Zealand pastoral hill country", aimed to find the best-fit coefficients of the Root Bundle Model with the Weibull survival function (RBMw), root distribution model (RDM), and root reinforcement model for their implementation in models such as BankforMAP and SlideforMAP. The authors have had much effort on experimenting such kind of research. This is an interesting topic, the results are used to formulate a general guideline for the planning of bio-engineering measures considering the temporal dynamics of poplar’s growth and their effectiveness in sediment and erosion control. However, there are some minor problems as indicated below. Finally, the presentation of the manuscript to reach the standard for a publication in an international science journal. The discussion also needs to be strengthen. Thus, I think that it will be published after minor revisions.
General comments
Comments for title:
Comment 1. “Populus nigra ita.”amend to " Populus nigra ita."
Comment 2. The manuscript is too long and there are too many figures and tables. It is suggested to appropriately delete unimportant figures and tables. In addition, the language expression of the full text can be properly deleted and condensed.
Comments for Abstract:
Comment 1. Line 17-24, There is some repetition in these sentences about the significance of the research results, and it is suggested to further condense and generalize.
Comments for Results:
Due to the length of the manuscript, it is suggested that the sentences in the result analysis and description sections should be cut to describe the important results.
Comments for figures and Tables:
Comment 1. For article length reasons, it is recommended to combine Figures 7, 21, and 22 into one figure.
Comment 2. Line 480, According to [24], this is a false expression.
NO
Author Response
Dear Reviewer,
Firstly, thank you very much for your clear and kind comments on the manuscript. We greatly appreciate your time dealing with the manuscript and your support.
Our responses to your comments and modifications of the manuscript have been stated in the attached PDF file.
We appreciate all your constructive suggestions. We are in the process of submitting the revised manuscript based on your comments. Thank you very much.
Best regards,
On behalf of the co-authors

Reviewer 3 Report (New Reviewer)
I have finished my review on the Manuscript Number: forests-2431385 Title: Analysis of poplar’s (Populus nigra ita.) root systems for quantifying bio-engineering measures in New Zealand pastoral hill country.
1. Generally, the manuscript presents an interesting topic.
2. There are some incomprehensible sentences. Please check again.
3. Authors must rewrite Abstract section. Especially in the materials and methods section.
4. The manuscript is well structured in Introduction section. However, it would be good to add some works about effects of vegetation on slope stability in line 36-37. Please read and add references as follows.
Sanandam Bordoloi, Charles Wang Wai Ng. The effects of vegetation traits and their stability functions in bio-engineered slopes: A perspective review. Engineering Geology. Volume 275, 20 September 2020, 105742.
R. Naghdi, S. Maleki……. Assessing the effect of Alnus roots on hillslope stability in order to use in soil bioengineering. JOURNAL OF FOREST SCIENCE, 59, 2013 (11): 417–423.
Misagh Parhizkar, Mahmood Shabanpour…… Evaluating the effects of forest tree species on rill detachment capacity in a semi-arid environment. Ecological Engineering 161 (2021) 106158.
Pooja Naredl, S Sangeetha. A Study on the Influence of Vegetation Growth on Slope Stability. IOP Conf. Series: Earth and Environmental Science. 1032 (2022) 012003.
Gang Huang, Mingxin Zheng, Jing Peng, "Effect of Vegetation Roots on the Threshold of Slope Instability Induced by Rainfall and Runoff", Geofluids, vol. 2021, Article ID 6682113, 19 pages, 2021.
5. The methodology is acceptable, and it doesn’t need to be rewritten.
6. The quality of Figure 4 is inappropriate.
7. The results and Discussion sections are OK.
Minor editing of English language required
Author Response
Dear Reviewer,
Firstly, thank you very much for your clear and kind comments on the manuscript. We greatly appreciate your time dealing with the manuscript and your support.
Our responses to your comments and modifications of the manuscript have been stated in the attached PDF file.
We appreciate all your constructive suggestions. We are in the process of submitting the revised manuscript based on your comments. Thank you very much.
Best regards,
On behalf of the co-authors

This manuscript is a resubmission of an earlier submission. The following is a list of the peer review reports and author responses from that submission.
Round 1
Reviewer 1 Report
This manuscript investigated the root distribution and reinforcement of poplar in New Zealand pastoral. Authors did a tough work, and this investigation will help to a better understanding of the control mechanism of soil erosion and sediment by root system. However, the results in this manuscript still needs reorganization. My main questions and concerns are about results and methods.
(1) The 23 figures and 11 tables in this manuscript are too complicated and tedious for readers. The most important thing should be the refinement and deletion of the manuscript. Please keep the core results which can support your conclusions.
(2) Line 147: “Living fine roots belonging to the sampled tree were counted …”. What method did you use to confirm the counted roots belonged to the sampled tree?
(3) Four trees for trenches excavation seems like the replication of this study, but authors exhibited the results for each tree separately rather than their average. Please explain the necessity.
(4) Line 248: 20% of the data were used for validation of the model. Please describe the method of data pick-out in detail.